

# Simulating oceanic radiocarbon with the FAMOUS GCM: implications for its use as a proxy for ventilation and carbon uptake

Jennifer E. Dentith[1], Ruza F. Ivanovic[1], Lauren J. Gregoire[1], Julia C. Tindall[1], Laura F. Robinson[2], and Paul J. Valdes[3]

[1]School of Earth and Environment, University of Leeds, Leeds, UK, LS2 9JT
[2]School of Earth Sciences, University of Bristol, Bristol, UK, BS8 1RJ
[3]School of Geographical Sciences, University of Bristol, Bristol, UK, BS8 1SS

*Correspondence to*: Jennifer E. Dentith (eejed@leeds.ac.uk)





**Abstract.** Constraining ocean circulation and its temporal variability is crucial for understanding changes in surface climate and the carbon cycle. Radiocarbon ($^{14}$C) is often used as a geochemical tracer of ocean circulation, but interpreting $\Delta^{14}$C in geological archives is complex. Isotope-enabled models enable us to directly compare simulated $\Delta^{14}$C values to $\Delta^{14}$C measurements and investigate plausible mechanisms for the observed signals. We have added three new tracers (water age, abiotic $^{14}$C, and biotic $^{14}$C) to the ocean component of the FAMOUS General Circulation Model to study large-scale ocean circulation and the marine carbon cycle. Following a 10,000 year spin-up, we prescribed the Suess effect (the isotopic imprint of anthropogenic fossil fuel burning) and the bomb pulse (the isotopic imprint of thermonuclear weapons testing) in a transient simulation spanning 1765 to 2000 CE. To validate the new isotope scheme, we compare the model output to direct $\Delta^{14}$C observations in the surface ocean (pre-bomb and post-bomb) and at depth (post-bomb only). We also compare the timing, shape and amplitude of the simulated marine bomb spike to $\Delta^{14}$C in geological archives from shallow-to-intermediate water depths across the North Atlantic. The model captures the large-scale structure and range of $\Delta^{14}$C values (both spatially and temporally) suggesting that, on the whole, the uptake and transport of $^{14}$C are well represented in FAMOUS. Differences between the simulated and observed values arise due to physical model biases (such as weak surface winds and over-deep North Atlantic Deep Water), demonstrating the potential of the $^{14}$C tracer as a sensitive, independent tuning diagnostic. We also examine the importance of the biological pump for deep ocean $^{14}$C concentrations and assess the extent to which $^{14}$C can be interpreted as a ventilation tracer. Comparing the simulated biotic and abiotic $\delta^{14}$C, we infer that biology has a spatially heterogeneous influence on $^{14}$C distributions in the surface ocean (between 18 and 30 ‰), but a near constant influence at depth (≈20 ‰). Nevertheless, the decoupling between the simulated water ages and the simulated $^{14}$C ages in FAMOUS demonstrates that interpreting proxy $\Delta^{14}$C measurements in terms of ventilation alone could lead to erroneous conclusions about palaeocean circulation. Specifically, our results suggest that $\Delta^{14}$C is only a faithful proxy for water age in regions with strong convection; elsewhere, the temperature dependence of the solubility of $CO_2$ in seawater complicates the signal.

# 1 Introduction

Understanding ocean circulation, how it has changed in the past, and how it might change in the future is crucial for understanding changes in surface climate and the carbon cycle (Rhein et al., 2013), but constraining ocean circulation is challenging due to the large range of spatiotemporal scales over which it operates (Talley, 2011). Modern methods for measuring ocean currents include moored arrays, drifting buoys, gliders, Acoustic Doppler Current Profilers, Argo floats, and satellite measurements (Dohan et al., 2010). Physical oceanographers also commonly infer ocean circulation from density distributions (Blanckenburg, 1999; Lynch-Stieglitz, 2001). Together, these techniques provide extensive spatiotemporal coverage in shallow and intermediate waters. However, historical and deep ocean measurements are much sparser (Rhein et al., 2013). The slowest ocean currents operate on centennial-to-millennial timescales (Talley, 2011), but there are no direct measurements of temperature and salinity, from which density is derived, or advection itself, beyond the instrumental record (i.e. prior to the late 1950s; Rhein et al., 2013). Instead, to determine past changes in ocean circulation, palaeoceanographers





rely on indirect ('proxy') measurements (e.g. carbon isotopes) in geological archives such as sediment cores and corals (Blanckenburg, 1999; Rahmstorf, 2002; Lynch-Stieglitz, 2003).

There are three naturally occurring carbon isotopes: the stable isotopes $^{12}$C (98.9 %) and $^{13}$C (1.1 %), and the radioactive isotope $^{14}$C ($1.2\times10^{-10}$ %), which is also known as radiocarbon (Key, 2001). Natural $^{14}$C is produced in the atmosphere by the cosmic spallation of nitrogen and enters the oceans via air-sea gas exchange. Once in the oceans, $^{14}$C is transported via large-scale ocean circulation and decays with a known half-life of 5730 years (Key, 2001). As there is no additional production of $^{14}$C in the interior ocean, the $^{14}$C content of deeper waters provides an indication of the time elapsed since the water was last in contact with the atmosphere (also termed the "water age" or "ventilation age"). Oceanic $^{14}$C distributions are also affected to a lesser extent by changes in atmospheric production (on multi-millennial timescales; Damon et al., 1978) and mass dependent fractionation during carbon cycle processes: air-sea gas exchange (e.g. Zhang et al., 1995), photosynthesis (e.g. Popp et al., 1989), and calcium carbonate formation (e.g. Emrich et al., 1970). Oceanographic $^{14}$C data are typically reported as $\Delta^{14}$C in per mil (‰) units (Stuiver and Polach, 1977), which is the $^{14}$C/$^{12}$C ratio of a sample relative to a standard, with corrections applied to account for fractionation effects and to normalise all samples relative to the mean value of terrestrial wood (Broecker and Walton, 1959; Key, 2001).

Between 1945 and 1963, thermonuclear weapons testing approximately doubled the amount of $^{14}$C in the atmosphere (Mahadevan, 2001). This artificial ("bomb") $^{14}$C spike has since been cycled through natural systems and can be utilised as a physical and biogeochemical tracer in terrestrial and marine settings (Scourse et al., 2012). For example, the flux, distribution and inventory of bomb $^{14}$C in the oceans can be used to constrain rates of air-sea gas exchange (Sweeney et al., 2007) and exchange between the shallow and deep ocean (Graven et al., 2012a). This is particularly valuable as both of these processes are important controls of oceanic $CO_2$ uptake (Graven et al., 2012a). The oceans are estimated to have absorbed approximately one third of anthropogenic $CO_2$ emissions, with an uncertainty of ±20 % (Khatiwala et al., 2013). More precise quantification of the size of the oceanic carbon sink is hindered by a lack of data from the sub-surface ocean, and the temporal duration and resolution of hydrographic measurements (Khatiwala et al., 2013).

A number of oceanographic surveys have been conducted since the early 1970s, providing an indication of large-scale carbon isotope distributions in the modern oceans that can be used to estimate the bomb $^{14}$C inventory and anthropogenic carbon uptake. These include: the Geochemical Ocean Sections Study (GEOSECS; 1972 to 1978; Östlund et al., 1988), Transient Tracers in the Ocean (TTO; 1981 to 1983; Brewer et al., 1985, 1986; Östlund and Grall, 1987), the South Atlantic Ventilation Experiment (SAVE; 1987 to 1989; Scripps Institute of Oceanography, 1992a, 1992b), Indian Gaz Ocean (INDIGO; 1985 to 1987; Sepanski, 1991), and the World Ocean Circulation Experiment (WOCE; 1990 to 1998; Orsi and Whitworth III, 2005; Talley, 2007; Koltermann et al., 2011; Talley, 2013). Although they provide a wealth of important data, one of the main shortcomings of these surveys is their low sampling frequency, with repeat measurements typically taken decades apart (Hood, 2009). They are therefore unable to capture the precise timing and amplitude of the bomb pulse. An additional limitation is that there are very few baseline (i.e. pre-bomb) measurements (especially in intermediate and deep waters) to contextualise anthropogenic changes relative to natural variability.





Isotopic ratios in geological archives (e.g. corals and bivalves) can complement direct oceanographic [14]C measurements (e.g. Sherwood et al., 2008; Scourse et al., 2012) and extend the record further back in time (e.g. Robinson et al., 2005; Burke and Robinson, 2012; Chen et al., 2015). Corals grow at a large range of depths (ranging from near-surface to more than 3000 m; Etnoyer and Morgan, 2005; Roark et al., 2009) for hundreds to thousands of years (Adkins et al., 2004;

Roark et al., 2009), with their carbonate skeletons recording the [14]C content of the seawater in which they formed (Druffel, 1980; Adkins et al., 2002; Farmer et al., 2015). Typically, warm-water corals grow in regions where the water temperature is between 23 °C and 29 °C (i.e. within 30° latitude of the equator and at <100 m depth; Spalding and Brown, 2015). Cold-water corals are found at shallow-to-intermediate depths (50 to 1000 m) in the high latitudes and at depths of up to 4000 m in the low latitudes, where the water temperature is between 4 °C and 12 °C (Roberts et al., 2006). Certain cold-water species (e.g.

bamboo corals) possess a two-part skeleton that can be used to simultaneously reconstruct both the surface and deep water [14]C signature. The proteinaceous nodes are formed from recently exported particulate organic matter, which reflects the isotopic signature of the surface ocean, whilst the calcite skeleton is derived from ambient dissolved inorganic carbon (Sherwood et al., 2008; Farmer et al., 2015). The difference between the node and skeleton $\Delta^{14}$C therefore provides an indication of the extent of vertical mixing in the water column (Sherwood et al., 2008). In contrast, most marine bivalve species are relatively

short lived (< 20 years) and have a narrow geographical range, which limits their utility for providing similarly long-term reconstructions (Weidman, 1995). However, "long-lived" species such as *Arctica islandica* have been widely used for generating more recent (i.e. 20[th] century) data sets (Scourse et al., 2012). *A. islandica* inhabit the continental shelves and slopes of the North Atlantic (between 10 and 200 m water depth and between 35° N and 70° N; Weidman, 1995). They commonly live for around 100 years and deposit carbonate shells with annual periodicity (Weidman, 1995).

Interpreting the [14]C signal in proxy records in terms of ocean circulation and marine biogeochemistry is complex. However, by simulating [14]C in numerical climate models we can directly compare the model output to $\Delta^{14}$C measurements and provide plausible mechanisms for the observed signals. Radiocarbon is not routinely incorporated into climate models because of the computational expense associated with fully spinning up the deep ocean and marine carbon cycle (Bardin et al., 2014). However, since the Ocean Carbon-Cycle Model Intercomparison Project (OCMIP) produced a legacy of standard input fields

and simulation setups (Orr et al., 2000), [14]C has been implemented into models of varying complexities, including: the UVic Earth System Model (Meissner et al., 2003; Koeve et al., 2015), the Hamburg LSG ocean circulation model (Butzin et al., 2005), MoBidiC (Crucifix, 2005), the Bern3D Earth System Model of Intermediate Complexity (Müller et al., 2006; Roth and Joos, 2013), CM2Mc (Galbraith et al., 2011), and CESM (Jahn et al., 2015). It is valuable to add another model, the FAMOUS General Circulation Model (GCM), to this list, not only to increase the Earth System capabilities of the model itself, but also

because examining the inter-model differences in $\Delta^{14}$C distributions can help us to better understand the underlying processes. Within the list of [14]C-enabled models, FAMOUS has the unique capability of being computationally efficient enough to fully spin-up the deep ocean circulation and the marine carbon cycle in a timely manner (without the need for offline or accelerated spin-up techniques), whilst still being able to maintain sufficient detail in the representation of the feedbacks between Earth System processes to study changes on decadal-to-centennial timescales. An additional benefit of adding [14]C to numerical





climate models is that it can be used to diagnose model biases. Temperature and salinity are commonly tuned to observations (Williamson et al., 2017), but $^{14}$C provides an independent constraint against which the simulated ocean circulation and marine carbon cycle can be evaluated.

Here, we describe the implementation of three new tracers in the ocean component of the FAMOUS GCM: water age, abiotic $^{14}$C, and biotic $^{14}$C (Section 2). The two representations of $^{14}$C differ in that the abiotic tracer is only affected by air-sea gas exchange, advection, and radioactive decay, whilst the biotic tracer is additionally cycled through the biological pump and is subject to isotopic fractionation during air-sea gas exchange and photosynthesis. We evaluate the performance of the model in simulating pre- and post-bomb $\Delta^{14}$C values by comparing the biotic tracer to observations (Sections 3.1 and 3.2, respectively). Specifically, the pre-bomb surface ocean $\Delta^{14}$C and post-bomb deep ocean $\Delta^{14}$C (i.e. natural $^{14}$C distributions)

are used to validate the model's large-scale ocean circulation and air-sea gas exchange scheme. The post-bomb surface ocean $\Delta^{14}$C (i.e. anthropogenic $^{14}$C) is used to further test the accuracy of the air-sea gas exchange, as well as vertical mixing in the shallow-to-intermediate water column. We also examine the transient bomb signal in natural archives (corals and bivalves) and in the model (Section 3.3). We compare the timing, magnitude, and shape of the $^{14}$C peak in different locations and at different water depths across the North Atlantic, and consider the implications for anthropogenic carbon uptake. Lastly, we

assess the extent to which $^{14}$C can be interpreted as a ventilation tracer. We consider the importance of the biological pump for deep water $^{14}$C concentrations by comparing the simulated biotic and abiotic $\delta^{14}$C distributions (Section 3.4). In this study, we use the term "biological pump" to refer to dissolved inorganic carbon (DIC) being converted into particulate organic carbon (POC) during photosynthesis, the transport of the POC from the shallow ocean into the abyssal ocean via gravitational settling, and its consequent remineralisation at depth, which is analogous to the "soft-tissue pump" outlined by Volk and Hoffert (1985).

We then assess how well the simulated biotic $^{14}$C ages compare to idealised water ages, which directly count the length of time since a water parcel was last in the uppermost layer of the ocean, and consider the implications for interpreting $\Delta^{14}$C proxy records from different regions (Section 3.5).

## 2 Methods

### 2.1 Model description

We have added three new tracers (water age, abiotic $^{14}$C, and biotic $^{14}$C) to the ocean component of the FAMOUS atmosphere-ocean GCM (Jones et al., 2005; Smith et al., 2008; Smith, 2012; Williams et al., 2013), which is a low resolution model derived from HadCM3 (Gordon et al., 2000; Pope et al., 2000). Briefly, the primitive equation atmospheric model has a horizontal resolution of 5° × 7.5° and 11 vertical levels on a hybrid sigma-pressure grid. The rigid-lid ocean is 2.5° × 3.75° with 20 vertical levels that vary in thickness from 10 m at the surface to more than 600 m at depth. The atmosphere operates

on a 1-h timestep, the ocean has a 12-h timestep, and the two components are coupled every 24-h. At the time of this study, and with a full-suite of ocean tracer fields enabled, FAMOUS is capable of simulating approximately 400 model years per





wallclock day on 16 core processors at the University of Leeds. It is therefore well suited to running the multi-millennial length simulations that are required to spin up deep ocean circulation and the marine carbon cycle.

In this study, we used the Met Office Surface Exchange Scheme (MOSES) version 1 (Cox et al., 1999) generation of the model. Although the MOSES2.2 generation of the model offers increased and more dynamic Earth System capabilities

(Essery et al., 2001, 2003; Williams et al., 2013; Valdes et al., 2017), the published setup does not accurately simulate the Meridional Overturning Circulation in multi-millennial simulations with constant pre-industrial boundary conditions (Dentith et al., 2019). At present, FAMOUS-MOSES1 is therefore a more appropriate tool for studying oceanic tracers. Nevertheless, our code is directly transferable between the different generations of FAMOUS and the parent model. The new carbon isotope scheme can therefore be implemented into FAMOUS-MOSES2.2 once the large-scale ocean circulation has been recalibrated,

or into HadCM3 for higher resolution scientific application.

The Hadley Centre Ocean Carbon Cycle (HadOCC) model is embedded within the ocean component of FAMOUS. In brief, HadOCC simulates air-sea gas exchange, the circulation of DIC, and the cycling of carbon by marine biota (Palmer, 1998; Palmer and Totterdell, 2001). Nutrients, phytoplankton, zooplankton, detritus, DIC, and alkalinity are simulated explicitly. The four biological components are considered in terms of their nitrogen contents, with the carbon contents and

fluxes calculated using fixed stoichiometric ratios. All six tracers are advected, diffused and mixed across all levels, although phytoplankton and zooplankton concentrations are negligible outside of the uppermost 100 m of the ocean. The primary mechanism for vertical carbon export is via detrital sinking, however, there is no representation of sediments. The small flux of detrital material that reaches the seafloor is therefore immediately refluxed back into the surface layer to conserve nitrogen and carbon. A more detailed description of HadOCC is provided by Dentith et al. (submitted).

## 20  2.2 Tracer implementation

### 2.2.1 Water age tracer

Carbon isotope ratios in oceanic geological archives, and $\Delta^{14}C$ in particular, are often interpreted in terms of water age (e.g. Stuiver et al., 1983; Broecker et al., 1990). To test this interpretation, we included a simple water age tracer in the model following the approach of England (1995). The water age tracer counts the number of timesteps since the water in a

single grid cell was last in contact with the atmosphere:

$$Age_{(t+\Delta t)} = Age_{(t)} + \Delta t \tag{1}$$

where $t$ is the current timestep.

The water age is instantly reset to zero in the surface layer, regardless of surface water residence times and whether or not air-sea gas exchange occurs (e.g. due to the presence of sea ice). This is therefore a highly idealised calculation, based

purely on physical ocean circulation, which can be used as a first-order comparative tool to understand the processes that influence $^{14}C$ age (such as air-sea gas exchange and ocean carbon cycle interactions) and how to interpret carbon isotope records of past ocean circulation.





### 2.2.2 Abiotic $^{14}$C

Abiotic $^{14}$C (i.e. $^{14}$C that is not affected by biological activity or isotopic fractionation) has previously been implemented into the ocean component of FAMOUS (Palmer, 1998). However, as the model has been further developed, this legacy code had not been maintained. At the outset of this study, initial tests revealed numerical instabilities (with $\Delta^{14}$C values

in excess of $\pm$1E6 ‰ developing during the first hundred years of the simulation), which eventually caused the model to crash. We therefore implemented an upper bound on the oceanic $CO_2$ flux (capping $pCO_2$ values at 1000 p.p.m. and the $CO_2$ flux at 30 mol m$^{-2}$ yr$^{-1}$). As a reference for future studies, we provide full documentation of the abiotic $^{14}$C implementation, which follows the OCMIP-2 protocol (Orr et al., 2000) with the following differences:

- We assume that modelled DIC is $^{12}$C and carry $^{14}$C as a ratio (DI$^{14}$C/DI$^{12}$C), therefore virtual fluxes are not required to
account for the dilution or concentration effects of surface freshwater fluxes (Appendix A).

- We carry $^{14}$C in model units to minimise the error associated with carrying small numbers:

$$Model\ units = \frac{DI^{14}C}{DI^{12}C} \times \frac{100}{{}^{14}C/{}_{{}^{12}C_{std}}} \qquad (2)$$

where $^{14}$C/$^{12}$C$_{std}$ = 1.176$\times$10$^{-12}$ (Karlen et al., 1965).

- In the calculation of the partial pressure of $CO_2$, we do not scale the mean ocean alkalinity with sea surface salinity. Instead
we continue to calculate the sea surface alkalinity using the standard equations in HadOCC, which consider the nutrient fluxes between the different organic pools and the rate of CaCO$_3$ production.

- In the calculation of aqueous $CO_2$, we use the carbonic acid constants of Roy et al. (1993) as opposed to Millero (1995) because this is consistent with the formulation of $CO_2$ solubility used in other areas of the model.

- In the calculation of the piston velocity, we use a coefficient of 0.31 cm h$^{-1}$ (Wanninkhof, 1992) instead of the 0.337 cm
h$^{-1}$ specified in the protocol. We also use the squared 10 m wind speed from the coupled atmospheric model as opposed to the climatology of the squared monthly mean of the instantaneous Special Sensor Microwave Imager (SSMI) velocity plus its variance. This approach is consistent with the gas transfer formulation used in other areas of the model and previous carbon isotope implementations in coupled atmosphere-ocean models (e.g. Jahn et al., 2015).

The air–sea gas flux of DI$^{12}$C ($F$) is calculated as:

$$F = PV \times (C_{sat} - C_{surf}) \qquad (3)$$

where $C_{sat}$ is the saturation concentration of atmospheric $CO_2$ (in mol m$^{-3}$), $C_{surf}$ is the surface aqueous concentration of $CO_2$ (in mol m$^{-3}$), and $PV$ is the piston velocity (in cm h$^{-1}$), which is calculated as:

$$PV = a \times u^2 \times (1 - a_{ice}) \times \left(\frac{660}{Sc}\right)^{-0.5} \qquad (4)$$

where $a$ is a tuneable coefficient, $u$ is the wind speed (in m s$^{-1}$), $a_{ice}$ is the fractional ice cover and $Sc$ is the Schmidt number
for $CO_2$, calculated as a function of sea surface temperature ($SST$, in °C):

$$Sc = 2073.1 - 125.62 \times SST + 3.6276 \times SST^2 - 0.043219 \times SST^3. \qquad (5)$$



The air–sea gas flux of $DI^{14}C/DI^{12}C$ $\left( F_{\frac{14}{12}} \right)$ is therefore calculated as:

$$F_{\frac{14}{12}} = \frac{1}{^{12}C} \times PV \times C_{sat} \times \left( \frac{^{14}A}{^{12}A} - \frac{^{14}C}{^{12}C} \right) \tag{6}$$

where $^{14}A/^{12}A$ and $^{14}C/^{12}C$ are the $^{14}C/^{12}C$ ratios of the atmosphere and DIC, respectively (Appendix B.1). Atmospheric $CO_2$ and $\Delta^{14}C$ concentrations can either be held constant or prescribed from a file that contains a single global weighted-average

value per year.

### 2.2.3 Biotic $^{14}C$

We implemented biotic $^{14}C$ as a ratio ($DI^{14}C/DI^{12}C$) in model units (Eq. (2)) following the same methodology used for $^{13}C$ (Dentith et al., submitted). We account for kinetic and equilibrium fractionation during air-sea gas exchange (Appendix B.2) based on the equations of Zhang et al. (1995), and calculate fractionation during photosynthesis as a function of aqueous

$CO_2$ concentration ($CO_2^*$) using the parameterisation of Popp et al. (1989). We do not account for fractionation during calcium carbonate formation because its inclusion has a negligible effect on the isotope distributions (Dentith et al., submitted). For all processes, the isotopic enrichment factor (ε) for $^{14}C$ is twice that of $^{13}C$ (ε14 = 2 × ε13; Craig, 1954), with

$$\varepsilon = (\alpha - 1) \times 1000 \ . \tag{7}$$

Biotic $^{14}C$ is also subject to radioactive decay in all four carbon pools, whereas $^{13}C$ is not.

### 2.2.4 Isotopic fractionation correction

To compare the simulated biotic $^{14}C$ values to observations, we apply the isotopic fractionation correction of Stuiver and Polach (1977):

$$\Delta^{14}C = \delta^{14}C - 2 \times \left( \delta^{13}C + 25 \right) \times \left( 1 + \frac{\delta^{14}C}{1000} \right) \tag{8}$$

where

$$\delta^{X}C = \left( \frac{^{X}C/_{^{12}C_{sample}}}{^{X}C/_{^{12}C_{standard}}} - 1 \right) \times 1000. \tag{9}$$

The "2" accounts for the mass dependency of isotopic fractionation and the "25" normalises all samples to the mean value for terrestrial wood (Key, 2001). As $\delta^{13}C$ is close to zero, the $\Delta^{14}C$ values are reduced by a near-constant value of -50 ‰ relative to the $\delta^{14}C$ values.

Other modelling studies typically compare their abiotic $\delta^{14}C$ values directly to $\Delta^{14}C$ observations because, without isotopic

fractionation effects, $\Delta^{14}C = \delta^{14}C$ (e.g. Toggweiler et al., 1989). In the absence of a biotic implementation, abiotic $\Delta^{14}C$ is a useful first-order representation of the processes that are important for the distribution of oceanic $\Delta^{14}C$ (air-sea gas exchange, advection, and radioactive decay). When both an abiotic and a biotic formulation are included in the same model, however, the value of comparing the two tracers is to examine the differences between the simulated fields to improve our understanding





of the processes that are important for the distribution of oceanic $^{14}$C. If the biotic $\Delta^{14}$C values are compared to the abiotic $\Delta^{14}$C values, the relationship between the two tracers is artifically reversed because the magnitude of the biotic fractionation correction (≈50 ‰) is larger than the uncorrected difference between the two tracers (which, as will be discussed in Section 3.4, is approximately 20 ‰). Consequently, we only compare the biotic $\Delta^{14}$C values to observations (Sections 3.1, 3.2, and 3.3) because they are a more complete representation of reality than the abiotic values. To assess the importance of the biological pump to the vertical profile of $^{14}$C, both in the global ocean and regionally, we compare the biotic and the abiotic tracers as uncorrected $\delta^{14}$C (Section 3.4).

In Section 3.5, the simulated biotic $^{14}$C concentrations (‰) are converted to $^{14}$C ages (relative to 1950 CE) as per Stuiver and Polach (1977):

$$^{14}C_{age} = -\frac{5730}{\ln(2)} \times ln\left(1 + \frac{\Delta^{14}C}{1000}\right). \tag{10}$$

### 2.2.5 Advection

Radiocarbon concentrations in the ocean interior are calculated as a function of 3-dimensional tracer transport and radioactive decay:

$$\frac{d\left(\frac{DI^{14}C}{DI^{12}C}\right)}{dt} = L\left(\left[\frac{DI^{14}C}{DI^{12}C}\right]\right) - \beta \times \frac{DI^{14}C}{DI^{12}C} \tag{11}$$

where $\beta$ is the radioactive decay constant (3.88915E-12 s$^{-1}$), which is based on a half-life of 5730 years (Godwin, 1962), and $L$ is the advection term. Flux-limited Quadratic Upstream Interpolation for Convective Kinematics (QUICK) advection (Leonard et al., 1993) is the default transport scheme in FAMOUS because it is positivity preserving and offers a better balance between numerical stability and diffusion compared to the standard alternatives (upstream differencing and centred differencing). It is used to transport all of the existing oceanic tracers (including temperature, salinity, and nutrients) and, for consistency, we have selected the same option for all three of our new tracers (water age, abiotic $^{14}$C, and biotic $^{14}$C).

### 2.3 Simulations

### 2.3.1 Spin-up simulation

We ran a 10,000 year spin-up simulation with constant pre-industrial boundary conditions to allow the deep ocean circulation and ocean carbon cycle to reach steady state. $\Delta^{14}$C$_{atm}$ was fixed at 0 ‰, $\delta^{14}$C$_{ocn}$ was initialised at a globally uniform value of 0 ‰ (i.e. biotic $\Delta^{14}$C$_{ocn}$ was initialised at -50 ‰), and the water age tracer was initalised at a globally uniform value of 0 years. The global volume-weighted integral of $\Delta^{14}$C started to stabilise after 6000 years and the water age stabilised after 8000 years (Figure S1). At the end of the spin-up simulation, the $\Delta^{14}$C drift was less than 0.001 ‰ yr$^{-1}$ (equivalent to a change in $^{14}$C age of 8.27 years per millennia), satisfying the OCMIP-2 criterion for steady state (Orr et al., 2000).



### 2.3.2 Historical simulation

We initialised a transient simulation for the period 1765 to 2000 CE from the end of the spin-up simulation to generate model output that is directly comparable to modern observations. Again, we followed the OCMIP-2 protocol with minor adjustments where necessary. The OCMIP-2 files contain biannual atmospheric $CO_2$ values and annual $\Delta^{14}C_{atm}$ values that are

separated into three latitude bands (90 to 20° N, 20° N to 20° S, and 20 to 90° S; Orr et al., 2000). However, as FAMOUS currently only allows a single atmospheric $CO_2$ value to be prescribed per model year, we calculated annual mean atmospheric $CO_2$ values from the OCMIP-2 data. At present, our isotope implementation also does not allow for latitudinal variability in $\Delta^{14}C_{atm}$. We therefore prescribed weighted global mean $\Delta^{14}C_{atm}$ values, which only differ from the regional values between 1956 and 1969 (Figure 1). The depletion of $\Delta^{14}C_{atm}$ to negative values during the early half of the 20th century is due to the

input of $^{14}C$-free $CO_2$ into the atmosphere from the burning of fossil fuels, known as the Suess effect (Suess, 1955; Keeling, 1979). Nuclear weapons testing commenced in 1945, and intensified between 1955 and 1963, rapidly enriching the atmosphere in $^{14}C$ (Mahadevan, 2001). The subsequent $\Delta^{14}C_{atm}$ decline primarily reflects the penetration of $^{14}CO_2$ into the oceans and the terrestrial biosphere (e.g. Graven et al., 2012a). To act as a control, the spin-up simulation was continued for an additional 235 years with constant atmospheric $CO_2$ and $\Delta^{14}C$.

### 3 Results and discussion

### 3.1 Pre-bomb surface ocean $\Delta^{14}C$ distributions

To assess the model performance in simulating natural (pre-bomb) $\Delta^{14}C$ distributions, we compare the simulated mean surface ocean $\Delta^{14}C$ values for the period 1955 to 1959 CE with historical surface measurements compiled by Graven et al. (2012b). We define this period as pre-bomb because, although atmospheric nuclear weapons testing intensified from 1955

onwards, the timescale for isotopic equilibration between the surface ocean and the atmosphere is 5 to 10 years (Toggweiler et al., 1989; Lynch-Stieglitz, 2003; Sarmiento and Gruber, 2006). The signature of bomb $^{14}C$ at the sea surface should therefore be minimal in the 5 year period immediately following its injection into the atmosphere. For example, Broecker and Walton (1959) estimated that the concentration of tropospheric $^{14}CO_2$ in the Northern Hemisphere increased by approximately 5 % per year between March 1955 and March 1958, but that only 10 % of the bomb $^{14}C$ produced in this period had entered the

oceans at the time of their study. Their calculations suggested that if this bomb $^{14}C$ was concentrated in the uppermost 100 m of the ocean, the mean surface ocean $\Delta^{14}C$ value in 1959 would be 12 to 32 ‰ higher than in 1955. In agreement with these calculations, between 1955 and 1960, the prescribed change in $\Delta^{14}C_{atm}$ is 210 ‰, but the simulated change in the globally averaged $\Delta^{14}C$ in the upper 100 m of the ocean is just 17 ‰. For comparison, the total simulated change (pre-bomb to peak bomb) in the upper 100 m of the ocean is 170 ‰. Therefore, whilst neither the simulated nor the observed values for this period

represent entirely natural $^{14}C$, these data provide a good indication of pre-bomb $\Delta^{14}C$ distributions in the absence of earlier (pre-1955) ship measurements. We focus our analysis on the surface ocean (0 to 10 m) because direct measurements of pre-





bomb $\Delta^{14}$C in the intermediate and deep ocean are scarce. There are 67 data points from the air-sea interface in the compilation of Graven et al. (2012b), however, there are only a further 125 data points at depths of 10 m and below (which accounts for more than 99.5 % of the global ocean volume). Binning the data according to the vertical levels in the model demonstrates that very few of these data points are at comparable depths (Table S1) and there are only a handful of locations that have multiple

measurements throughout the entire water column. These data are therefore insufficient for providing a coherent picture of natural $\Delta^{14}$C distributions in the deep ocean.

        The model captures the overall structure of the observations in the surface ocean (Figure 2a). The highest values ($\approx$ -52 ‰) are in the sub-tropical gyres where surface water residence times are relatively long, allowing for greater equilibration with the atmosphere. Equatorial regions have intermediate $\Delta^{14}$C values due to the combined effect of older ($^{14}$C-depleted)

waters from the deep ocean upwelling back to the sea surface and weaker winds than in the sub-tropics reducing the input of $^{14}$C from the atmosphere. The lowest values ($\approx$ -150 ‰) are in the high latitudes because (1) sea ice inhibits air-sea gas exchange, (2) surface water residence times are relatively short, and (3) older water is mixed upwards from the abyssal ocean to the surface ocean at sites of deep water formation. In absolute terms, the simulated values are, on average, 60 ‰ lower than observed in the Pacific Ocean and 30 ‰ lower than observed in the Atlantic Ocean (Figure 2b). The discrepancy may be partly

reconciled by the envelope of uncertainty on the pre-bomb observations (5 to 36 ‰; Graven et al., 2012b), which encompasses much of the offset between the simulated and observed $\Delta^{14}$C values in the Atlantic basin. However, this does not account for the larger bias in the Pacific basin, which is better explained by the atmospheric forcing, specifically the timing and geographical distribution of early nuclear weapons testing. Approximately 70 atmospheric nuclear weapons tests were conducted by the U.S.A., U.S.S.R. and U.K. between 1945 and 1955 in Kazakhstan, Nevada, and the Pacific Ocean (Yang et

al., 2003). The Pacific observations could therefore include a small bomb signal from the preliminary testing that took place on the Bikini and Enewetak Atolls. The model does not capture this early period of nuclear activity because we prescribed a globally uniform bomb signal beginning in 1955 (Sect. 2.3.2).

### 3.2 Post-bomb $\Delta^{14}$C distributions

        To assess the model performance in the post-bomb era, we compare the simulated mean $\Delta^{14}$C values for the 1990s

with data from version 1 of the Global Data Analysis Project (GLODAP; Key et al., 2004), which is a 3-dimensional compilation of measurements from approximately 12,000 hydrographic stations. Overall, the model shows good agreement with the observations, with a global linear regression $r^2$ value 0.75 and a root mean square error (RMSE) of 55 ‰ (Figure 3). The best agreement is in the Pacific Ocean (where the $r^2$ value is 0.86 and the RMSE is 57 ‰) and the worst agreement is in the Southern Ocean (where the $r^2$ value is 0.52 and the RMSE is 52 ‰).

At the sea surface, the model successfully replicates the large-scale distribution of $\Delta^{14}$C, with the highest values in the sub-tropics, intermediate values in the tropics, and the lowest values in the polar regions (Figure 4 and Figure 5). This demonstrates that, to the first order, the processes that control the uptake and transport of $^{14}$C (air-sea gas exchange, biological activity, and ocean circulation) are well represented in the model. By investigating the reasons for some of the differences



between the simulated and observed values in more detail, we can assess the relative importance of each of these processes in controlling the $\Delta^{14}C$ distributions and establish how well the isotope scheme captures the behaviour of the model. For example, the simulated values are too high southwards of $\approx 30°$ S due to a combination of simulation set-up and model biases. Specifically, prescribing a latitudinally-uniform atmospheric $\Delta^{14}C$ (Sect. 2.3.2) means that the input value is between 2 and

206 ‰ too high in the Southern Hemisphere between 1956 and 1969 (Figure 1). Consequently, the influx of $^{14}C$ into the surface ocean in this region is too large. This is accentuated by insufficient sea ice being simulated in the Southern Ocean, promoting excessive air-sea gas exchange. Furthermore, the simulated surface winds are weaker than observed (e.g. Kalnay et al., 1996). Although this reduces the input of $^{14}C$ from the atmosphere into the surface ocean (opposing the effects of the high input value and insufficient sea ice), it also results in a relatively shallow mixed layer. Reduced vertical mixing of the surface signal

therefore leads to an accumulation of $^{14}C$ in the uppermost layers of the ocean. Conversely, but for similar reasons, the simulated values are lower than observed in the Northern Hemisphere sub-tropical gyres, largely because the input of $^{14}C$ is between 2 and 187 ‰ too low northwards of 20° N (Figure 1). The simulated values are also lower than observed in the eastern equatorial regions due to the excessive upwelling of $^{14}C$-depleted waters from the deep ocean (Palmer and Totterdell, 2001).

At depth, the highest $\Delta^{14}C$ values (youngest waters) are simulated in the Atlantic Ocean, with intermediate values in

the Indian Ocean, and the lowest values (oldest waters) in the Pacific Ocean, in agreement with observations (Figure 6 and Figure S2). The bomb signal (positive $\Delta^{14}C$ values) and newly formed deep waters (higher $\Delta^{14}C$ values) are clearly identifiable in the zonal means (Figure 6). However, the penetration of the bomb signal (white and red colours in Figure 6) is a few hundred metres too shallow in the model. This is due to the aforementioned insufficient convective mixing, which results in a relatively shallow mixed layer and excessive pooling of $^{14}C$ in the upper ocean. The simulated $\Delta^{14}C$ values also indicate that the abyssal

Atlantic waters are too well ventilated as a result of over-deep North Atlantic Deep Water (NADW) formation and insufficient Atlantic-sector Antarctic Bottom Water (AABW) formation, which are known limitations of the FAMOUS GCM (Smith, 2012; Dentith et al., 2019). Across the whole ocean, the observed minimum $\Delta^{14}C$ value is -240 ‰, which occurs in the northeast North Pacific Ocean, at a depth of approximately 2500 m. In the model, there is weak ($< 1$ Sv) convection to around 3 km depth in the sub-polar North Pacific (Dentith et al., 2019), which prevents the accumulation of old, $^{12}C$-enriched (low $\Delta^{14}C$)

waters. Instead, the simulated $\Delta^{14}C$ minimum (-215 ‰) is in the eastern equatorial Pacific at a depth of approximately 1500 m. Thus, the discrepancies between the simulated and observed $\Delta^{14}C$ distributions demonstrate that the $^{14}C$ isotope scheme is capturing the physical behaviour of the model well.

To examine the model performance in more detail, we have sub-divided the global ocean into 14 regions of interest, which include the sub-tropical gyres, deep water formation regions (simulated and/or observed), upwelling zones, and common

coral locations (Figure 7). The simulated and observed depth profiles are well-matched, both regionally and globally (Figure 8), further supporting the notion that (on the whole) the uptake and transport of $^{14}C$ are well represented in the model. The globally-averaged simulated $\Delta^{14}C$ values are a near-perfect match to the observed values at depths below 2500 m and, interestingly, there is excellent agreement between the simulated and observed $\Delta^{14}C$ values in the Southern Hemisphere deep water formation region (SH_DWF) and the Southern Ocean upwelling zone (SO_UP). We therefore infer that, although the





Antarctic Circumpolar Current in FAMOUS is weak compared to observations (Dentith et al., 2019), it is still strong enough to homogenize the water column.

As previously discussed, and similar to other $^{14}$C-enabled models (e.g. Jahn et al., 2015), many of the differences between the simulated and observed $\Delta^{14}$C distributions can be explained by known physical biases. For example, the simulated

$\Delta^{14}$C gradient between the surface ocean and approximately 1000 m depth is shallower than observed because of insufficient convective mixing. This is visible in all regions, except the Northern Hemisphere deep water formation region (NH_DWF) and the North Pacific (NP), where convection in the model is deeper than it should be (Dentith et al., 2019), thereby actively mixing the bomb signal into the sub-surface waters. In the modern oceans, NADW is formed in the Labrador and Nordic Seas (Kuhlbrodt et al., 2007); therefore, the observed depth profiles in these regions (LS and NH_DWF, respectively) are very

similar. However, FAMOUS does not simulate deep water formation in the Labrador Sea, so the model has a much shallower $\Delta^{14}$C profile here. In each of the observed Northern Hemisphere profiles (LS, NH_ASG, NH_DWF, NP, and NS), there is a negative $\Delta^{14}$C excursion from $^{14}$C-depleted Antarctic Intermediate Water between 1000 and 1500 m depth. This water mass is not represented in FAMOUS, so the simulated depth profiles are relatively smooth. Similarly, in the Southern Hemisphere Atlantic sub-tropical gyre (SH_ASG), the positive excursion in the $\Delta^{14}$C measurements between 1500 m and 3000 m reflects

the influx of $^{14}$C-enriched NADW. In FAMOUS, the positive excursion extends below 4000 m because, as previously discussed, the modelled NADW cell has a greater vertical range than observed. The model accurately replicates the deep ocean values in the equatorial upwelling zones (EEA_UP and EEP_UP), which means that the waters being mixed upwards from the abyssal ocean towards the sea surface have approximately the correct isotopic signature. However, the strong upwelling rates in FAMOUS (Palmer and Totterdell, 2001) create an offset between the simulated values and the $\Delta^{14}$C measurements in the

shallow and intermediate waters. The masking in the GLODAP data set also contributes towards some of the offset between the model and the observations. For example, we include the relatively low Arctic Ocean values in our global, Northern Hemisphere deep water formation region (NH_DWF) and Labrador Sea (LS) profiles, but these latitudes are masked out in GLODAP due to the sparsity of data (Figure 4a). Overall, the regional depth profiles corroborate the skill of the biotic $^{14}$C scheme in correctly capturing the physical behaviour of the model and demonstrate the potential of the new tracer for providing

an independent constraint for future recalibration work (e.g. to improve the representation of the Atlantic Meridional Overturning Circulation in FAMOUS).

### 3.3 Comparison to natural archives

To better understand the penetration of the bomb signal into the ocean, we compare the transient surface and shallow-to-intermediate water $\Delta^{14}$C values in our model with coral and bivalve records from 12 sites across the North Atlantic (Figure 9).

Collectively, these archives span the period between the late 1800s and the early 2000s, thereby providing a record of pre-bomb $\Delta^{14}$C, the timing and magnitude of peak $\Delta^{14}$C values, and the subsequent rate of decline. We present 16 published records in total (Table S2). The 7 bivalve records (as presented in the compilation of Scourse et al., 2012) are all from the uppermost 100 m of the water column. Five of the corals are from intermediate depths (between 362.5 m and 1410.0 m). Three of these





corals are bamboo corals, which simultaneously record the surface and ambient $\Delta^{14}C$ (Section 1). We have also included a surface coral record from Bermuda to complement the deep-sea record from this site.

Bomb $^{14}C$ is observed in all 16 published records and in the corresponding model output (Figure 10). The peak oceanic values are consistently lower than peak atmospheric values (Figure 1) because of the relative size of the two carbon pools

(Ciais et al., 2013). In general, the model captures both the relative timing and the overall shape of the observed profiles very well. This reaffirms the skill of FAMOUS in representing carbon uptake and transport. It also suggests that large-scale processes (such as air-sea gas exchange and vertical mixing) are more important for determining the manifestation of the marine bomb pulse than local processes (such as riverine input and exchange between coastal basins and the open ocean), which are not represented as accurately in the model.

In the shallow ocean (0 to 100 m), the ambient $\Delta^{14}C$ profiles (both simulated and observed) closely resemble the simulated surface $\Delta^{14}C$ profiles (Figure 10), demonstrating how efficiently the bomb signal is transferred throughout the mixed layer. Bomb $^{14}C$ is detected at all sites almost immediately following its injection into the atmosphere, with the simulated surface ocean $\Delta^{14}C$ values starting to increase as early as 1956 and 1957. At every site, the rate of increase from pre-bomb to peak $\Delta^{14}C$ values is faster than the rate of decline, indicating that air-sea gas exchange is more efficient than vertical mixing

between shallow and intermediate waters. By the year 2000, all sites still have elevated $\Delta^{14}C$ relative to the natural levels.

Examining the similarities and differences between the simulated and observed timeseries, and between the simulated timeseries in different locations, can also help to improve our understanding of the important processes controlling the expression of the marine bomb spike, both in the model and in reality. As noted by Scourse et al. (2012), the hydrographic setting of each site influences the time taken for bomb $^{14}C$ to be detected, the overall strength of the signal, and its residence

time. The surface timeseries therefore fall into three categories: high amplitude-early peaks, low amplitude-late peaks, and intermediate amplitude peaks (Figure 10). In agreement with observations, the highest $\Delta\Delta^{14}C$ (peak $\Delta^{14}C$ minus pre-bomb $\Delta^{14}C$) values are simulated at Oyster Ground (OG; ≈370 ‰) and German Bight (GB; ≈365 ‰), with peak values attained in 1972 and 1969, respectively. These are both shallow, coastal sites that have small carbon reservoirs (Figure 9 and Figure 11), therefore they are strongly influenced by air-sea gas exchange. Conversely, the lowest $\Delta\Delta^{14}C$ values are simulated at

Siglufjörður (S; 70.0 ‰) and Grimsey (G; 70.0 ‰), where the $\Delta^{14}C$ values plateau between 1970 and 2000. These sites are in the NADW formation region, where the water column is well mixed, with $^{14}C$-depleted waters being upwelled from the abyssal ocean and $^{14}C$-enriched surface waters being quickly transported to depth (Figure 11). The simulated timeseries have higher variability at sites where convection is less persistent (GeB and NE; Figure 10) and similar variability is captured by the Oyster Ground (OG) bivalve, which is subject to increased stratification in the summer months (Scourse et al., 2012). The Tromsø

(T) bivalve is located in a fjord that is strongly influenced by the North Atlantic Current and the Norwegian Coastal current (Scourse et al., 2012), but this unique hydrographic setting is not captured by FAMOUS. Instead, Tromsø is within the model's Northern Hemisphere deep water formation region. The model therefore simulates an attenuated bomb peak relative to the observations (Figure 10), which is comparable to the simulated timeseries at other sites that are affected by persistent deep convection, such as Grimsey (G) and Siglufjörður (S). The observed surface ocean profiles from the Hudson Strait (HS) and



Grand Banks (GrB) corals are very similar (Figure 10) because both sites are influenced by the Labrador Current, which has a one year transient time from HS in the northwest to GrB in the southeast (Sherwood et al., 2008). FAMOUS does not simulate deep water formation in the Labrador Sea and coastal currents in this semi-enclosed region are not well resolved by the model. The simulated surface peaks at these sites are therefore of a higher amplitude than observed, more so at Grand Banks (GrB),

which is less affected by seasonal sea ice than the Hudson Strait (HS). It is interesting to note that, in the model, similar bomb profiles are simulated in very different hydrographic settings, for example in the Hudson Strait (HS) and Bermuda (B). The Hudson Strait (HS) is a semi-enclosed setting that is characterised by weak surface currents although, as previously discussed, it should also be influenced by deep convection (but is not in the model). In contrast, Bermuda (the furthest site from the coast included in this study) is influenced by strong horizontal advection (sub-tropical gyre circulation) and weak vertical mixing.

In general, both of the simulated profiles adequately capture the shape and timing of the observed surface timeseries, however, the two observed profiles differ in that the Hudson Strait (HS) could be classed as having a low-to-medium amplitude peak whilst Bermuda (B) has a medium-to-high amplitude peak.

As expected, in the intermediate ocean (362.5 m to 1410.0 m), the bomb signal is lagged and damped relative to the surface ocean (Figure 10). For example, in the Northeast Channel (NE), bomb $^{14}$C is detected in the ambient $\Delta^{14}$C signal in

1961 (5 years later than in the surface ocean), with peak values simulated in 1989 (14 years later than in the surface ocean). The simulated $\Delta\Delta^{14}$C at depth is ≈100 ‰ compared to ≈140 ‰ in the surface ocean, and by the year 2000, the ambient values had only decreased by ≈10 ‰ compared to ≈110 ‰ in the surface layer. Given the temporal resolution of the coral records, it is unclear whether the $\Delta^{14}$C values in the intermediate ocean have peaked at the end of the timeseries, but we can use the isotope-enabled model to predict the depth to which the bomb $^{14}$C has penetrated (Figure 11) and thus infer ongoing trends.

For example, the relatively high resolution Grand Banks (GrB) record still appears to be on an upward trajectory in the year 2000, which is corroborated by the model output (Figure 10). However, the 1410 m Bermuda (B) coral records a ≈5 ‰ decrease in $\Delta^{14}$C between 1999 and 2001 (Figure 10). Additional measurements would be needed to confirm whether this is natural variability or a permanent reversal, but because the model accurately captures the observed signal at this site, we infer that the $\Delta^{14}$C values at ≈1400 m have not peaked by the year 2000 (Figure 10 and Figure 11). We can also use the isotope-enabled

model to fill in the gaps when there is a lack of ambient data. For example, there are only three data points from intermediate water depths in the Hudson Strait (HS), each of which is within 15 ‰ of the nearest dated surface measurement (Figure 10). From these data alone, it is therefore unclear whether the $\Delta^{14}$C values at intermediate depths peak at a similar time to the surface ocean or whether the intermediate ocean responds more slowly. Again, however, we use the model to infer that peak $^{14}$C values have not been attained at ≈400 m depth by the year 2000. In fact, the model suggests that, by the year 2000, the

$\Delta^{14}$C values have only peaked at the shallowest of the intermediate ocean sites included in this study, the Northeast Channel (NE; Figure 10 and Figure 11).

The difference between the surface and ambient $\Delta^{14}$C can be used to infer the extent of vertical mixing in the water column (Sherwood et al., 2008). In both the Hudson Strait (HS) and Grand Banks (GrB) coral records, the intermediate water values are similar to the surface ocean values in the pre-bomb era, demonstrating that the water column is well mixed to depths



of at least 400 m and 700 m, respectively. In the Northeast Channel (NE), the average difference between the observed surface and ambient values in the pre-bomb era is approximately 15 ‰, suggesting that the water column is more stratified off the coast of Nova Scotia than it is further to the north (in the Labrador Sea) and east (off the coast of Newfoundland). The model simulates a larger difference (approximately 10 to 30 ‰) between the surface and ambient signal at all four of the sites where
the natural archives cover multiple depths (HS, GrB, NE, and B), which corroborates our earlier interpretation that the water column in FAMOUS is less well ventilated than observed.

Overall, this comparison demonstrates the utility of the isotope-enabled model for providing plausible data to fill in spatiotemporal gaps in proxy records and for corroborating suggestions from observational studies about the processes controlling the transfer of carbon from the atmosphere to shallow and intermediate water depths in different hydrographic
settings. It also underlines the skill of the isotope scheme in highlighting physical model biases (e.g. insufficient convection in the Labrador Sea), which could be improved by retuning the model.

### 3.4 Influence of the biological pump

To analyse the influence of biology on the $^{14}$C distributions in FAMOUS, it is useful to compare the simulated biotic and abiotic $\delta^{14}$C values (Eq. (9)) in the surface ocean and at depth. As outlined in Sections 2.2.3 and 2.2.2, respectively, the
biotic tracer is cycled through the biological pump and is subject to isotopic fractionation during air-sea gas exchange and photosynthesis, whereas the abiotic tracer is only affected by air-sea gas exchange, advection, and radioactive decay.

We infer that the biological pump is the dominant control on the differences between the biotic and abiotic $\delta^{14}$C values in the surface ocean because the spatial distributions of the $\delta^{14}$C difference closely resemble the spatial distributions of the simulated primary productivity. However, we acknowledge that fractionation during air-sea gas exchange is a secondary effect,
which will exacerbate the difference between the two tracers in regions of $CO_2$ outgassing, where $^{12}$C is preferentially released to the atmosphere, and reduces the difference between the two tracers in regions of $CO_2$ invasion, where $^{12}$C is preferentially taken up into the oceans. In the deep ocean, the influence of air-sea gas exchange is negligible, therefore we attribute the differences between the biotic and the abiotic schemes entirely to the biological pump.

As expected, the simulated biotic $\delta^{14}$C values are higher than the corresponding abiotic $\delta^{14}$C values everywhere in the
global surface ocean (Figure 12). This is because the biotic tracer accounts for the preferential uptake of $^{12}$C during primary productivity, which leaves the DIC pool relatively enriched in $^{14}$C, whereas the abiotic tracer is not affected by biological fractionation. In the pre-industrial ocean, the offset between the two tracers ranges between ≈18 ‰ in the Southern Hemisphere Pacific sub-tropical gyre (where the primary productivity is relatively low) and ≈29 ‰ in the productive equatorial upwelling zones, with a mean difference of ≈21 ‰. In the post-bomb era, the offset ranges between ≈20 ‰ and ≈35 ‰, with a mean
difference of ≈23 ‰. Notably, in both cases, the anomaly is larger in the eastern equatorial Atlantic Ocean than the eastern equatorial Pacific Ocean, even though the Pacific region has higher simulated primary productivity. We propose that this asymmetry relates to the age of the waters that are being upwelled in each basin. The upwelling Pacific waters are approximately 600 years older than water that is being upwelled from the deep Atlantic basin (Figure S3). The deep Pacific



waters therefore have a lower $\delta^{14}C$ signature as a result of radioactive decay. However, both the abiotic and biotic schemes account for this effect. Instead, we suggest that the primary cause of the asymmetrical difference between the biotic and abiotic tracers is that the older Pacific waters contain a larger proportion of remineralised organic matter, which is enriched in $^{12}C$ in the biotic scheme, reinforcing the lower $\delta^{14}C$ signal that is being mixed upwards into the surface waters.

We also assess the importance of the biological pump for transporting $^{14}C$ into the deep ocean (Figure S4 and Figure S5). We have focussed our analysis on the same 14 regions of interest outlined in Section 3.2 and Figure 7. In both the pre-industrial and the post-bomb ocean, the globally-averaged difference between the biotic and abiotic $\delta^{14}C$ in the deep ocean is approximately 19.7 ‰ (Figure 13). This is lower than the mean difference between the two tracers in the surface ocean because the remineralisation of $^{12}C$-enriched particulate organic carbon reduces the biotic $\delta^{14}C$ at depth, but the abiotic $\delta^{14}C$ is
unaffected by this process. The $\delta^{14}C$ difference in the eastern equatorial Pacific upwelling zone (EEP_UP) is 0.6 ‰ higher than the global mean difference in the pre-industrial ocean, and 0.75 ‰ higher in the post-bomb ocean. In each of the 13 other regions of interest, the deep ocean difference between the abiotic and biotic tracers is close to the global mean difference in both timeslices. $\Delta^{14}C$ measurements from proxy records are typically reported to 1 decimal place (e.g. Sherwood et al., 2008; Scourse et al., 2012) and the errors in the coral and bivalve data presented in Section 3.3 range between 2.1 ‰ and 22.0 ‰,
with an average error of approximately 5.5 ‰. The offset between the $\delta^{14}C$ difference in the eastern equatorial Pacific upwelling zone (EEP_UP) and the global mean difference (which is the largest spatial disparity; Figure 13) is therefore of the same order of magnitude as the precision of $^{14}C$ measurements and is well within the analytical error. Thus, we infer that, from an analytical perspective, the biological pump has a spatially constant influence on deep ocean $^{14}C$ concentrations, which could be accounted for with a global correction of approximately 20 ‰. Simulations performed with other $^{14}C$-enabled models
(Section 1) would be needed to verify how model-dependent our suggested correction is. Furthermore, sensitivity experiments would be required to verify whether the same conclusion (and correction) holds true for palaeo studies, for example, at the Last Glacial Maximum, when there is evidence that the spatial distribution and overall levels of primary productivity were different from present (although there is no overall consensus as to whether the biological pump was weaker or stronger; Shemesh et al., 1993; Kumar et al., 1995; Anderson et al., 1998).

**3.5 Comparison to water age**

Radiocarbon ages are commonly used as a proxy for the length of time since a water parcel was last in contact with the atmosphere (e.g. Stuiver et al., 1983; Broecker et al., 1990). To assess the validity of this interpretation, we compare the simulated $^{14}C$ ages (Eq. (10)) with the idealised water ages at the end of the 10,000 year spin-up simulation. By subsetting the data in two different ways, we are able to identify specific regions where the water ages are well represented by $^{14}C$ ages, as
well as regions where the relationship breaks down. Firstly, we consider the major ocean basins and divide the water column into shallow (0 to 550 m), intermediate (550 to 2500 m) and deep (2500 to 5500 m) water based on the components of overturning circulation described by Talley (1999). We also compare the water age and $^{14}C$ age depth profiles in the 14 regions of interest outlined in Figure 7.



As expected, the simulated $^{14}$C ages are consistently older than the simulated water ages in the surface ocean, where the water age is preconditioned to be zero. The $^{14}$C reservoir effect ranges between approximately 450 years in the sub-tropical gyres and 1300 years in the Southern Ocean (Figure 14a), reflecting the 5 to 10 year timescale required for isotopic equilibration between the ocean and the atmosphere (Toggweiler et al., 1989; Lynch-Stieglitz, 2003; Sarmiento and Gruber,

2006), which is significantly longer than surface water residence times (e.g. 2 years for Antarctic Surface Waters; Lynch-Stieglitz et al., 1995). Consequently, in the shallow ocean (0 to 550 m; red shapes in Figure 15), the lowest $r^2$ values are in the Southern Ocean because sea ice and short surface water residence times limit air-sea gas exchange, which increases the $^{14}$C reservoir ages (Figure 14b and Figure 14c). The convective mixing of $^{14}$C-depleted waters from the abyssal ocean into the shallow ocean at sites of deep water formation further increases the reservoir effect. The highest shallow ocean $r^2$ values are

in the Indian Ocean where relatively long surface water residence times allow the surface ocean to come closer to equilibrium with the atmosphere (reducing the $^{14}$C reservoir effect) and, in contrast to the Atlantic and Pacific basins, there is no significant upwelling of older ($^{14}$C-depleted) waters from the deep ocean.

The regional depth profiles demonstrate that, where there is convective mixing (i.e. in the near-surface ocean), the $^{14}$C ages and water ages follow similar patterns, with an offset due to the aforementioned incomplete air-sea gas exchange

(Figure 16 and Figure S6). Considering the water column as a whole, the water ages generally increase with depth because they are a simple function of advection. In contrast, the $^{14}$C ages typically decrease or remain near constant with depth below approximately 1000 m, as per the DIC concentrations. In Section 3.4, we concluded that the biological pump has a spatially constant influence on deep ocean $^{14}$C concentrations. We therefore infer that the shapes of the regional $^{14}$C age profiles are largely controlled by the solubility pump, which we can separate into two components: the physical component (i.e. ocean

circulation) and the chemical component (i.e. the temperature dependence of the solubility of $CO_2$ in seawater, with increased solubility in cold – high latitude and deep ocean – waters relative to warm – low latitude and surface ocean – waters). Our simulations demonstrate that $^{14}$C is a good tracer for water age in regions where the physical component is more dominant (i.e. in well mixed regions such as the Drake Passage, DP, the Northern Hemisphere deep water formation region, NH_DWF, and the Tasman Sea, TS; and in shallow marginal seas, such as the Caribbean Sea, CS, and the Labrador Sea, LS), and that

elsewhere, where the chemical component is more dominant, the relationship between water age and $^{14}$C age breaks down.

In the deep ocean (2500 to 5550 m; blue shapes in Figure 15), the $r^2$ values in the Atlantic, Pacific and Southern Oceans are all less than 0.1, demonstrating that, at basin-scale, the water age distributions at these depths are not well represented by the $^{14}$C ages (Figure 14f and Figure 14g). In contrast, the $r^2$ value in the deep Indian Ocean is 0.67. We propose two main reasons as to why the correlation between the water ages and the $^{14}$C ages in the deep Indian Ocean is higher than

elsewhere. Firstly, there is no deep water formation in this basin, which mixes young water with a relatively high $^{14}$C age into the abyssal Atlantic and Southern Oceans. Secondly, the average surface ocean temperature is between 4.5 °C and 18 °C higher than in the other basins, which means that the chemical component of the solubility pump is weaker.

Previous studies have also identified problems with using $^{14}$C to infer deep ocean ventilation rates. For example, Campin et al. (1999) implemented a water age tracer and abiotic $^{14}$C into a 3° × 3° ocean only GCM. In their model, the



simulated $^{14}$C ages of NADW and AABW were systematically older than the water ages, which the authors attributed to incomplete air-sea equilibration as a result of short surface water residence times with respect to air-sea gas exchange rates and interference from sea ice. The $^{14}$C ages of NADW and AABW in FAMOUS are >700 years and >1000 years older than the water ages, respectively (Figure 14). In agreement with Campin et al. (1999), we propose that this is largely due to the $^{14}$C

surface reservoir effect. However, Campin et al. (1999) did not account for isotopic fractionation in their study, nor was their $^{14}$C tracer cycled through the marine biological pump. Based on the results of our abiotic-biotic $\delta^{14}$C comparison (Section 3.4), we suggest that the biological pump reduces the $^{14}$C reservoir effect (more so in the surface ocean than at depth) because it enriches the DIC pool in the heavier isotope (Figure 13).

Overall, we have demonstrated that the simulated $^{14}$C distributions are sensitive to a mixture of physical and

biogeochemical processes, therefore, in agreement with Campin et al. (1999), we suggest that interpreting $^{14}$C ages in terms of ventilation alone may lead to erroneous conclusions about palaeocean circulation. Proxy data from the deep ocean are important for understanding how physical ocean circulation may have changed in the past, but our model results suggest that, to be interpreted in terms of water age, the records must be carefully selected from well mixed regions, and should be interpreted as a local not a basin-wide signal. The isotope-enabled model is therefore a useful tool for identifying plausible

convective regions in the geological past (for example under glacial boundary conditions), which could provide valuable data for improving our understanding of how circulation has changed through time.

## 4 Summary

We have added three new tracers (water age, abiotic $^{14}$C, and biotic $^{14}$C) to the ocean component of the FAMOUS GCM to study ocean circulation and the marine carbon cycle. The model accurately simulates large-scale $\Delta^{14}$C distributions

both spatiallly (in the surface ocean and at depth) and temporally (in the pre-bomb era and the post-bomb era), and is able to capture the timing, shape, and amplitude of the marine bomb pulse at various locations across the North Atlantic Ocean. This therefore suggests that, on the whole, the uptake and transport of $^{14}$C are well represented in FAMOUS. Differences between the simulated and observed values arise due to model biases, including weak surface winds, over-deep NADW, insufficient Atlantic-sector AABW formation, and an absence of deep convection in the Labrador Sea. This therefore demonstrates that

the new $\Delta^{14}$C tracer is accurately capturing the physical behaviour of the model and will be a useful tuning metric for recalibrating FAMOUS in the future, for example, to improve the representation of the Atlantic Meridional Overturning Circulation.

Radiocarbon ages in geological archives are typically interpreted in terms of ventilation. To assess the validity of this interpretation, we first examined the importance of the biological pump to deep ocean $^{14}$C concentrations by comparing the

simulated biotic and abiotic $\delta^{14}$C values. The biotic $\delta^{14}$C values are higher than the abiotic values at all depths because the biotic tracer accounts for the preferential uptake of $^{12}$C during primary productivity (whereas the abiotic tracer is not affected by biological fractionation) and remineralisation in the deep ocean occurs without further fractionation. In the surface ocean,





the difference between the two tracers reflects the large-scale patterns of primary productivity, but the difference is near constant at depth (both vertically and regionally). We therefore propose that this could be accounted for with a global correction of approximately 20 ‰. Secondly, we compared the simulated $^{14}$C ages with the idealised water ages. We conclude that, in our model, the water ages are not well represented by the $^{14}$C ages at global- or basin-scale, but $^{14}$C is a good tracer for water

5    age in well-mixed regions, where the physical component of the solubility pump is a more dominant control of DIC distributions than the chemical component. Due consideration of the balance between physical and biogeochemical processes should therefore be exercised when interpreting $\Delta^{14}$C in proxy records to avoid drawing erroneous conclusions about palaeocean circulation.

**Code availability**

10    The main repository for the Met Office Unified Model (UM) version 4.5, as presented in this study, can be found at http://cms.ncas.ac.uk/code_browsers/UM4.5/UMbrowser/index.html. The files required to add each of the new tracers are available via the Research Data Leeds Repository (https://doi.org/10.5518/621). The UM configuration ("basis") files for the simulations described in this paper can be accessed via the Providing Unified Model Access (PUMA) service (http://cms.ncas.ac.uk/wiki/PumaService).

**Table 1:** Overview of the simulations described in this study, as denoted by their unique five letter Met Office UM identifiers and the notation used within this manuscript.

| Identifier | Simulation | Duration |
|---|---|---|
| XOAVB | spin-up | 0 to 10,000 years |
| XOAVI | Transient | 1765 to 2000 CE |
| XOGNC | Control | 1765 to 2000 CE |

**Data availability**

The data are available via the Research Data Leeds Repository (https://doi.org/10.5518/621).

20    **Author contributions**

RFI designed and supervised the project. JED wrote and implemented the isotope code with input from JCT, LJG, and RFI. PJV wrote and implemented the idealised water age code. JED ran the simulations, analysed the results, and prepared the manuscript with input from all co-authors.



**Competing interests**

The authors declare that they have no conflict of interest.

**Acknowledgements**

JED was funded by the Natural Environment Research Council (NERC) SPHERES Doctoral Training Partnership (grant
number: NE/L002574/1). RFI acknowledges support from NERC grant NE/K008536/1. The contribution of JCT was
supported through the Centre for Environmental Modelling And Computation (CEMAC), University of Leeds. Numerical
climate model simulations made use of the N8 High Performance Computing (HPC) Centre of Excellence (N8 consortium and
EPSRC Grant #EP/K000225/1) and ARC2, part of the HPC facilities at the University of Leeds, UK. We thank Alexandra
Jahn (University of Colorado) for helpful discussions about implementing carbon isotopes in GCMs and Robin Smith
(University of Reading) for supplying the code to cap the oceanic $CO_2$ flux.

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





**Figures**

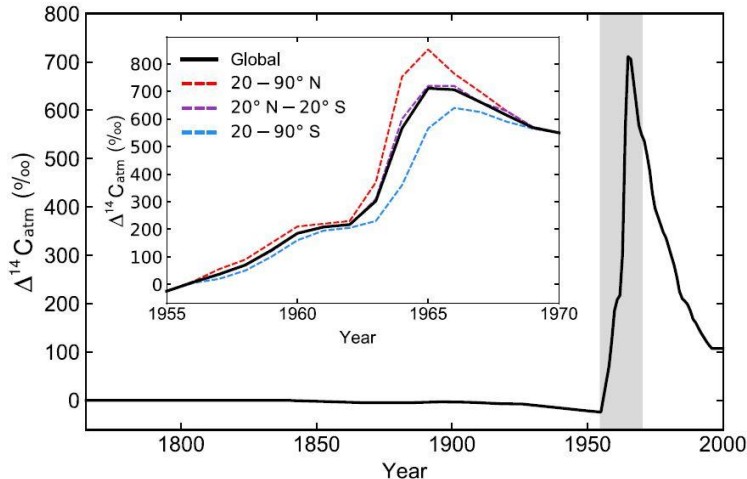

**Figure 1:** Prescribed atmospheric $\Delta^{14}$C values (1765 to 2000 CE). Inset (1955 to 1970, shaded): Weighted global mean (black, prescribed) compared to the three latitude bands outlined in the OCMIP-2 files (Orr et al., 2000); northern hemisphere (90 to 20° N, red), tropics (20° N to 20° S, purple), and southern hemisphere (20 to 90° S, blue).

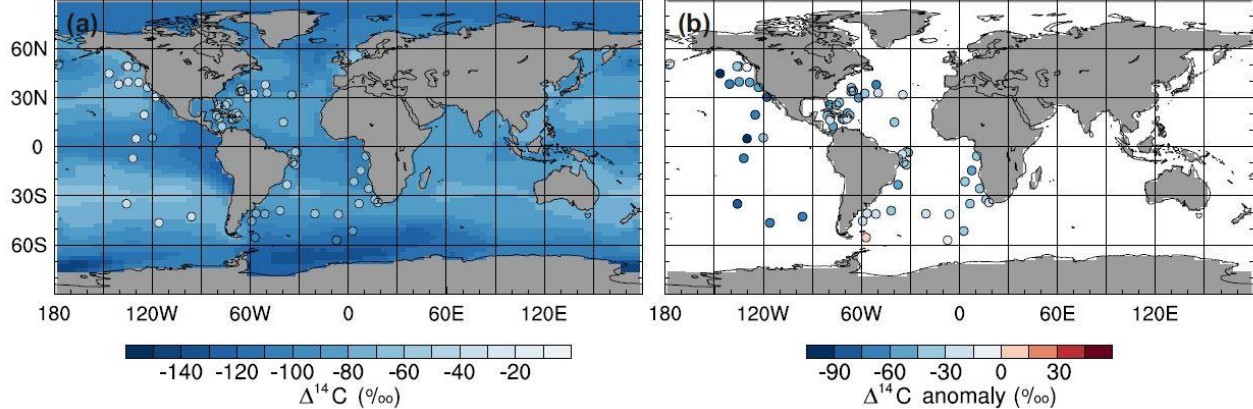

**Figure 2:** (a) Mean surface ocean $\Delta^{14}$C (1955 to 1959 CE; coloured contours) overlain with historical surface measurements (filled dots) for the same period (compiled by Graven et al., 2012b) and (b) simulated minus observed $\Delta^{14}$C.



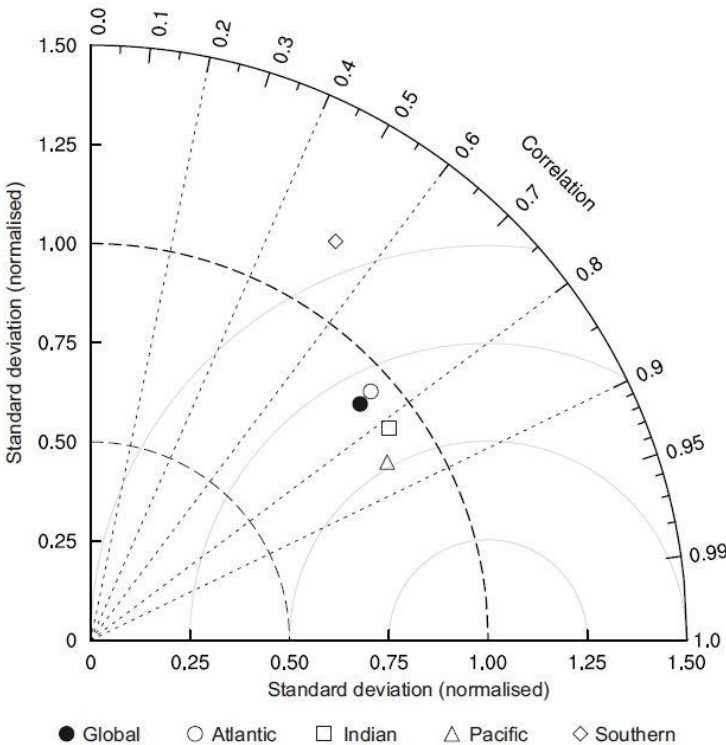

**Figure 3:** Taylor plot of simulated $\Delta^{14}C$ relative to the ungridded GLODAP observations from the 1990s (Key et al., 2004) separated by major ocean basin: global (filled circle), Atlantic (hollow circle), Indian (square), Pacific (triangle), and Southern (diamond). A perfect simulation would have a correlation coefficient of 1 and a normalised standard deviation (simulated standard deviation/observed standard deviation) of 1.





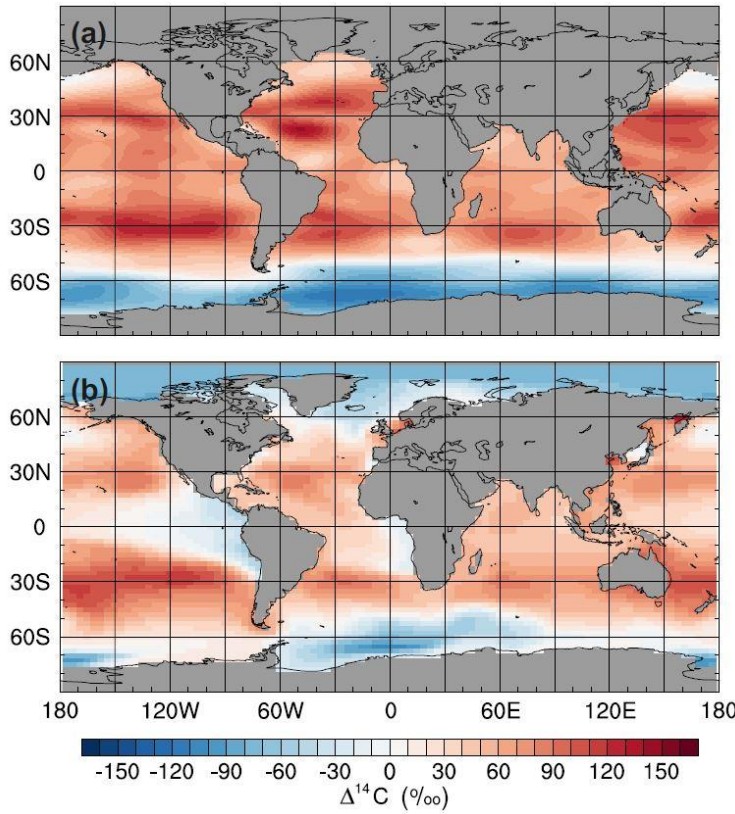

**Figure 4:** Mean surface ocean $\Delta^{14}$C during the 1990s: (a) the gridded GLODAP data (Key et al., 2004) and (b) the simulated values.



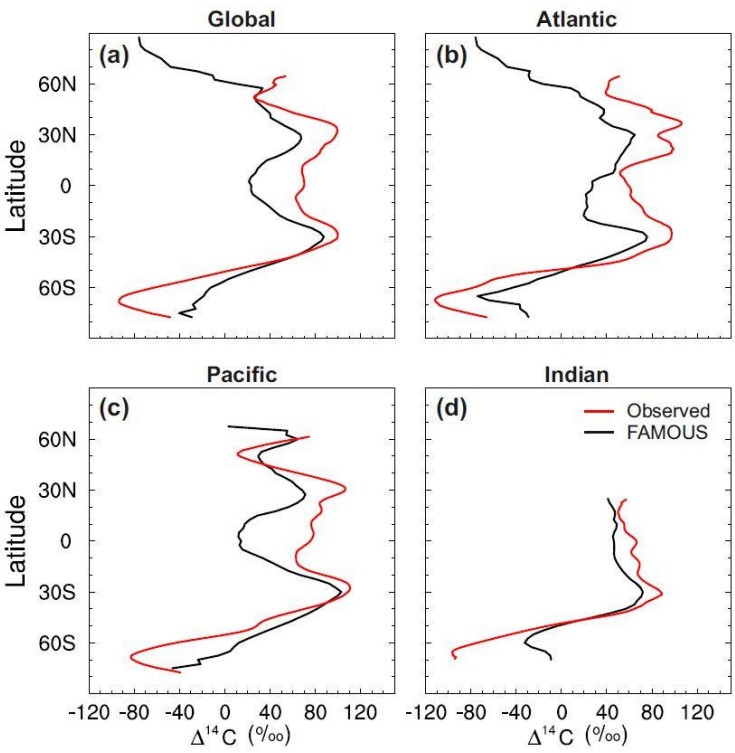

**Figure 5:** Zonal mean surface ocean $\Delta^{14}C$ during the 1990s: (a) global ocean, (b) Atlantic Ocean, (c) Pacific Ocean, (d) Indian Ocean.





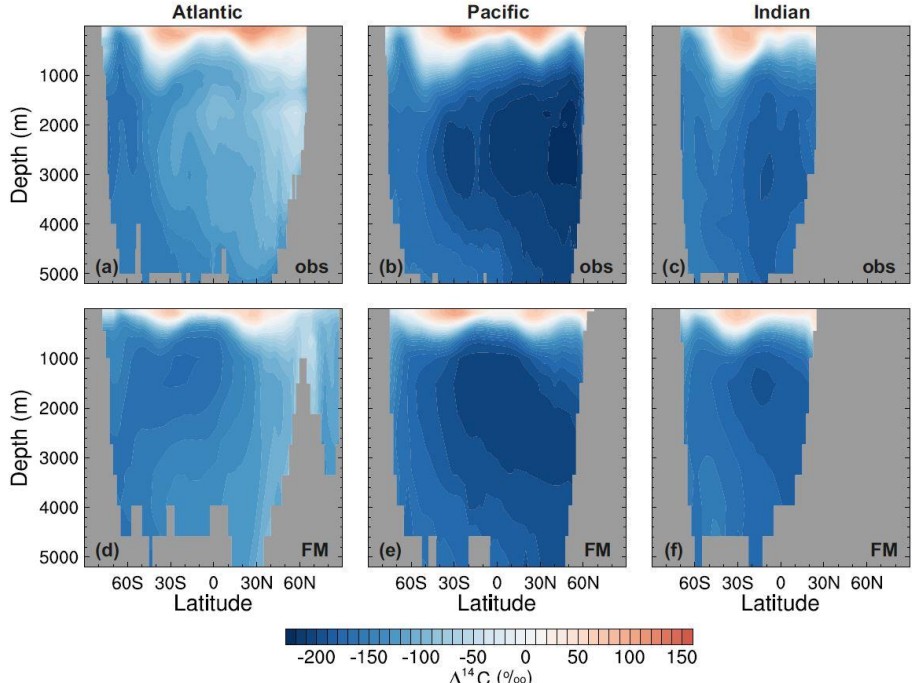

**Figure 6:** Zonal mean $\Delta^{14}$C during the 1990s in the Atlantic Ocean (left), Pacific Ocean (centre) and Indian Ocean (right): the gridded GLODAP data (Key et al., 2004; top) and the simulated values (bottom).



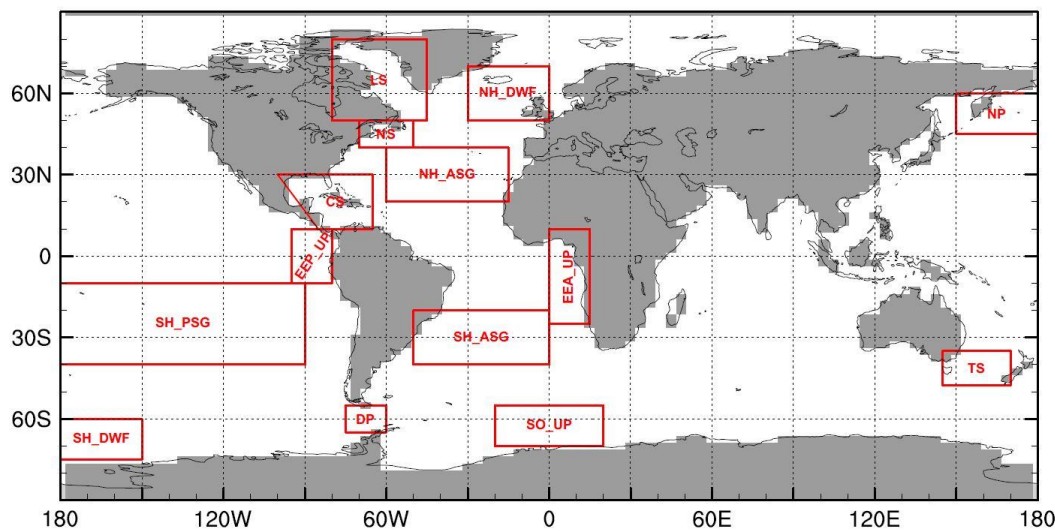

**Figure 7:** Location map of the 14 regions of interest: Caribbean Sea (CS), Drake Passage (DP), eastern equatorial Atlantic upwelling zone (EEA_UP), eastern equatorial Pacific upwelling zone (EEP_UP), Labrador Sea (LS), Northern Hemisphere Atlantic sub-tropical gyre (NH_ASG), Northern Hemisphere deep water formation region (NH_DWF), North Pacific (NP), Nova Scotia (NS), Southern Hemisphere Atlantic sub-tropical gyre (SH_ASG), Southern Hemisphere deep water formation region (SH_DWF), Southern Hemisphere Pacific sub-tropical gyre (SH_PSG), Southern Ocean upwelling zone (SO_UP), and Tasman Sea (TS).







**Figure 8:** Global (G) and regional depth profiles of simulated (black) and observed (red) $\Delta^{14}$C during the 1990s. The regions are outlined in Figure 7: Caribbean Sea (CS), Drake Passage (DP), eastern equatorial Atlantic upwelling zone (EEA_UP), eastern equatorial Pacific upwelling zone (EEP_UP), Labrador Sea (LS), Northern Hemisphere Atlantic sub-tropical gyre (NH_ASG), Northern Hemisphere deep water formation region (NH_DWF), North Pacific (NP), Nova Scotia (NS), Southern
5   Hemisphere Atlantic sub-tropical gyre (SH_ASG), Southern Hemisphere deep water formation region (SH_DWF), Southern Hemisphere Pacific sub-tropical gyre (SH_PSG), Southern Ocean upwelling zone (SO_UP), and Tasman Sea (TS).

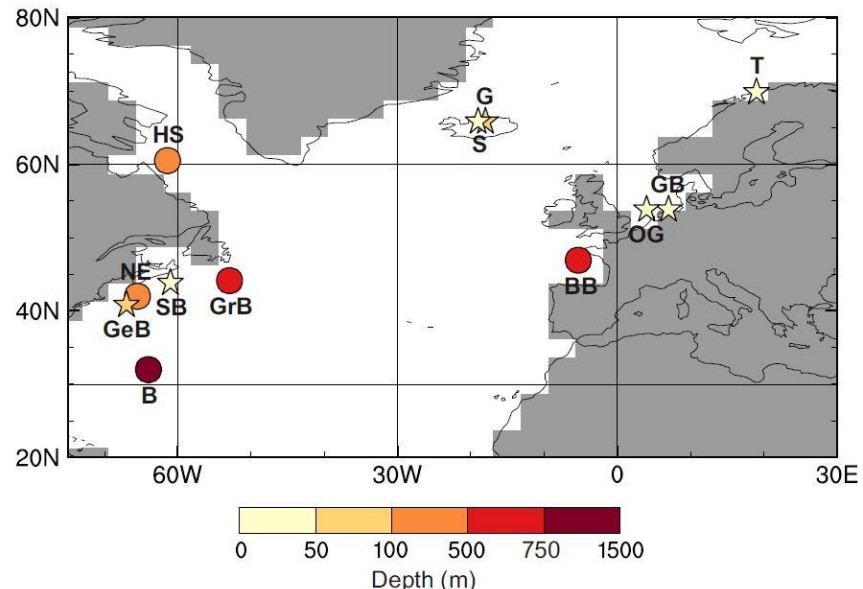

**Figure 9:** Location map of the North Atlantic coral (circles) and bivalve (stars) data used in this study: Bermuda (B), Bay of
10   Biscay (BB), Grimsey (G), German Bight (GB), Georges Bank (GeB), Grand Banks (GrB), Hudson Strait (HS), Northeast Channel (NE), Oyster Ground (OG), Siglufjörður (S), Sable Bank (SB), and Tromsø (T). The depth of the archive is denoted by the marker colour.



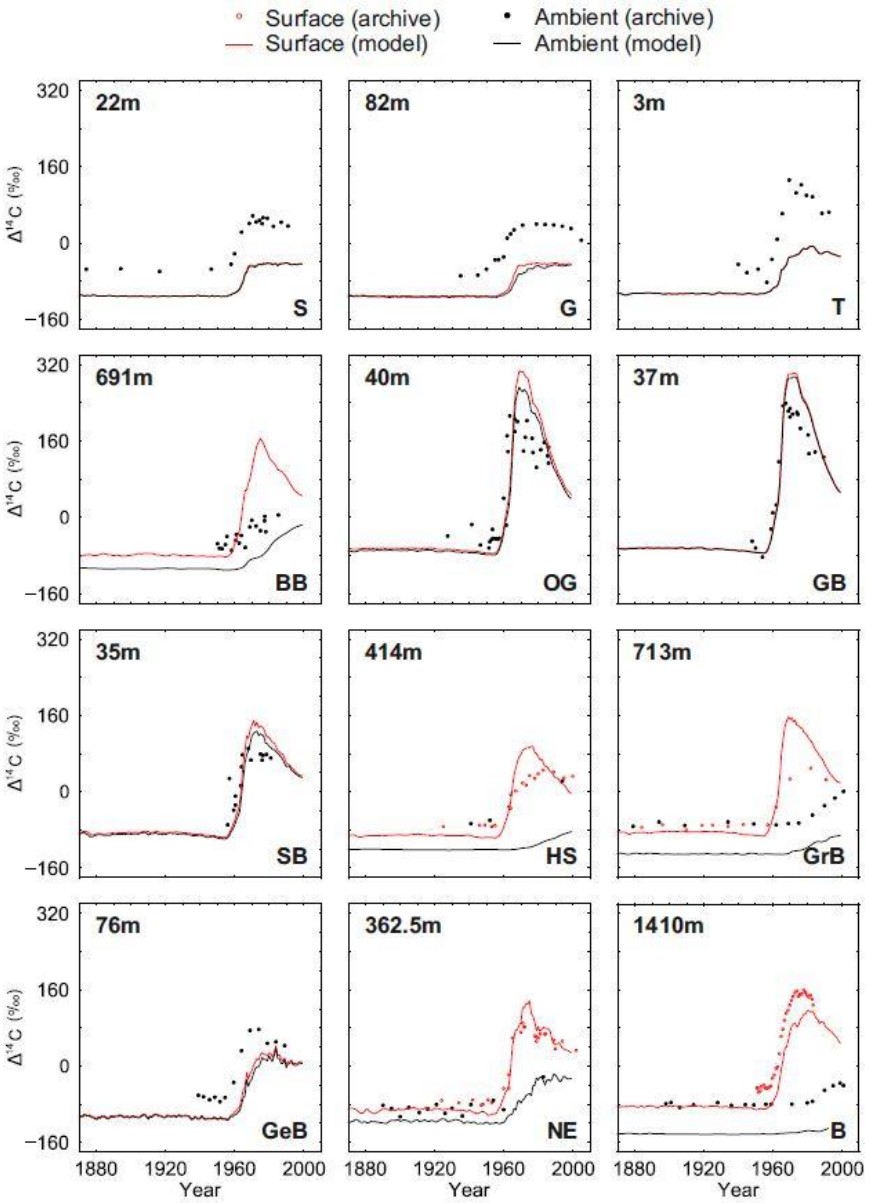

**Figure 10:** Simulated (lines) and observed (markers) surface (red) and ambient (black) $\Delta^{14}$C at the coral and bivalve locations (outlined in Figure 9): Bermuda (B), Bay of Biscay (BB), Grimsey (G), German Bight (GB), Georges Bank (GeB), Grand Banks (GrB), Hudson Strait (HS), Northeast Channel (NE), Oyster Ground (OG), Siglufjörður (S), Sable Bank (SB), and Tromsø (T). Note that only the bamboo corals (GrB, HS, and NE) and Bermuda (B) have surface observations. Additionally, the simulated surface and ambient $\Delta^{14}$C values are the same for Tromsø (T) because the surface layer in the model is 10 m deep.



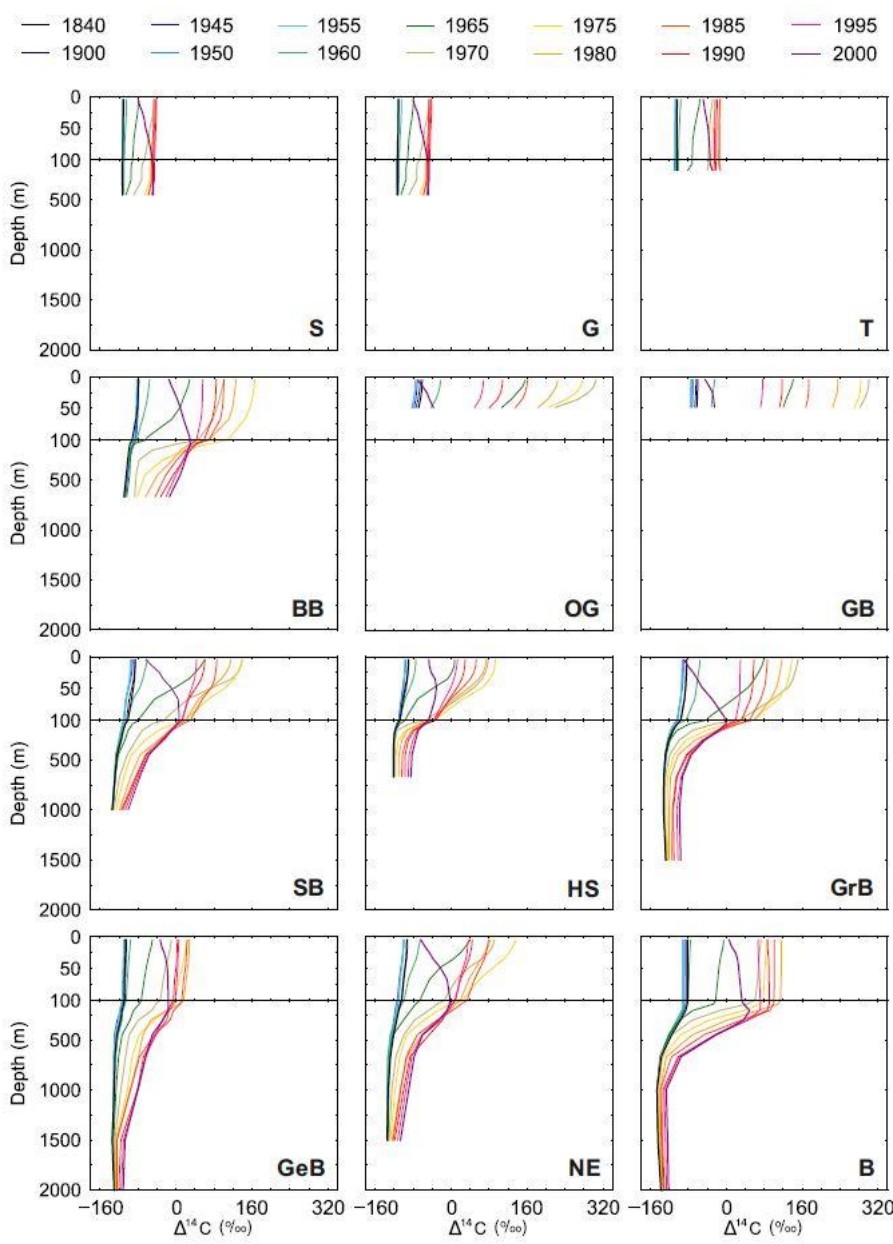

**Figure 11:** Transient depth profiles of simulated $\Delta^{14}$C at the coral and bivalve locations (outlined in Figure 9): Bermuda (B), Bay of Biscay (BB), Grimsey (G), German Bight (GB), Georges Bank (GeB), Grand Banks (GrB), Hudson Strait (HS), Northeast Channel (NE), Oyster Ground (OG), Siglufjörður (S), Sable Bank (SB), and Tromsø (T). Note that the vertical scale has been expanded for the uppermost 100 m of the water column.



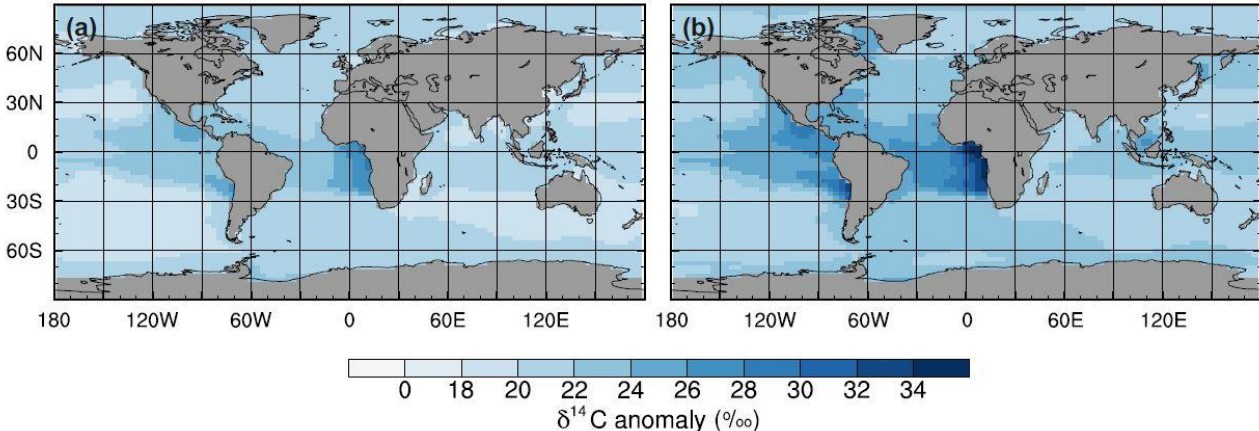

**Figure 12:** Biotic minus abiotic surface ocean $\delta^{14}$C: (a) the end of the spin-up simulation (years 9900 to 10,000) and (b) the 1990s.



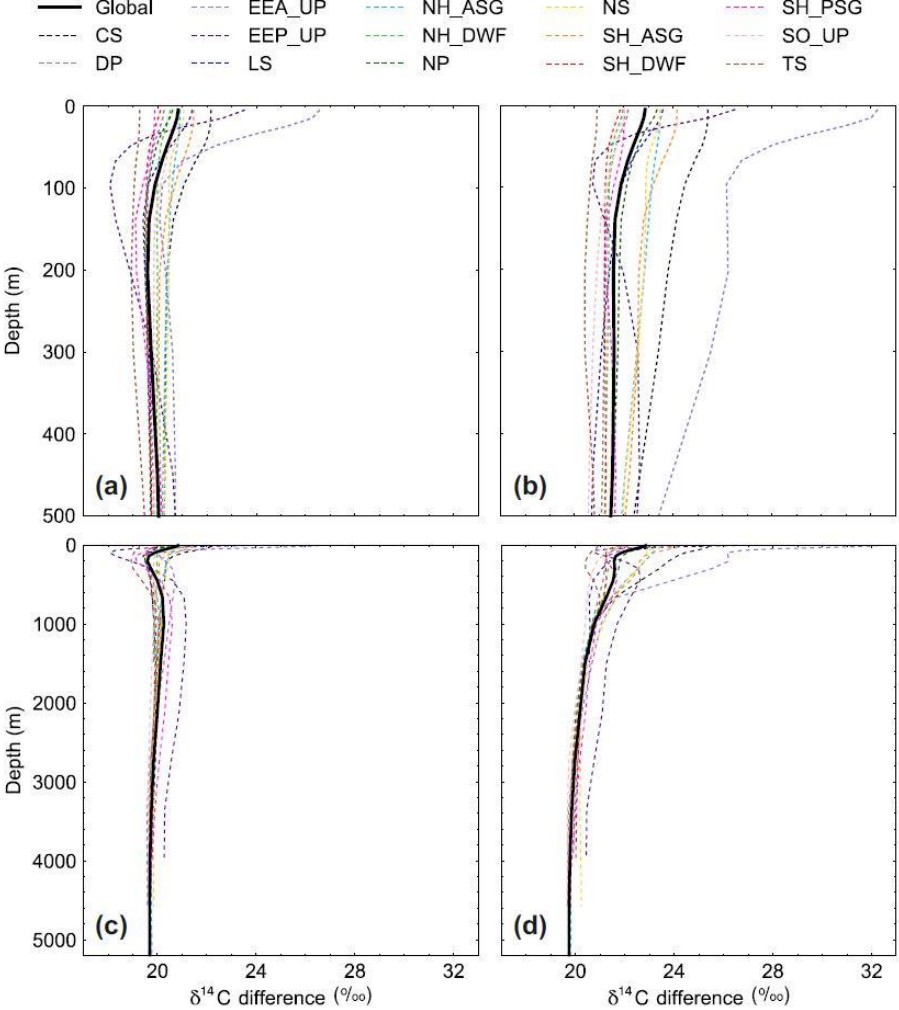

**Figure 13:** Global (solid black) and regional (dotted) depth profiles of biotic minus abiotic $\delta^{14}$C: (a, c) the end of the spin-up simulation (years 9900 to 10,000) and (b, d) the 1990s. The regions are outlined in Figure 7: Caribbean Sea (CS), Drake Passage (DP), eastern equatorial Atlantic upwelling zone (EEA_UP), eastern equatorial Pacific upwelling zone (EEP_UP), Labrador Sea (LS), Northern Hemisphere Atlantic sub-tropical gyre (NH_ASG), Northern Hemisphere deep water formation region (NH_DWF), North Pacific (NP), Nova Scotia (NS), Southern Hemisphere Atlantic sub-tropical gyre (SH_ASG), Southern Hemisphere deep water formation region (SH_DWF), Southern Hemisphere Pacific sub-tropical gyre (SH_PSG), Southern Ocean upwelling zone (SO_UP), and Tasman Sea (TS).





**Figure 14:** $^{14}$C ages (left) and water ages (right) at the end of the spin-up simulation (years 9900 to 10,000): (a) the surface ocean (0 to 10 m), (b – c) the shallow ocean (0 to 550 m), (d – e) the intermediate ocean (550 to 2500 m), and (f – g) the deep ocean (2500 to 5500 m).


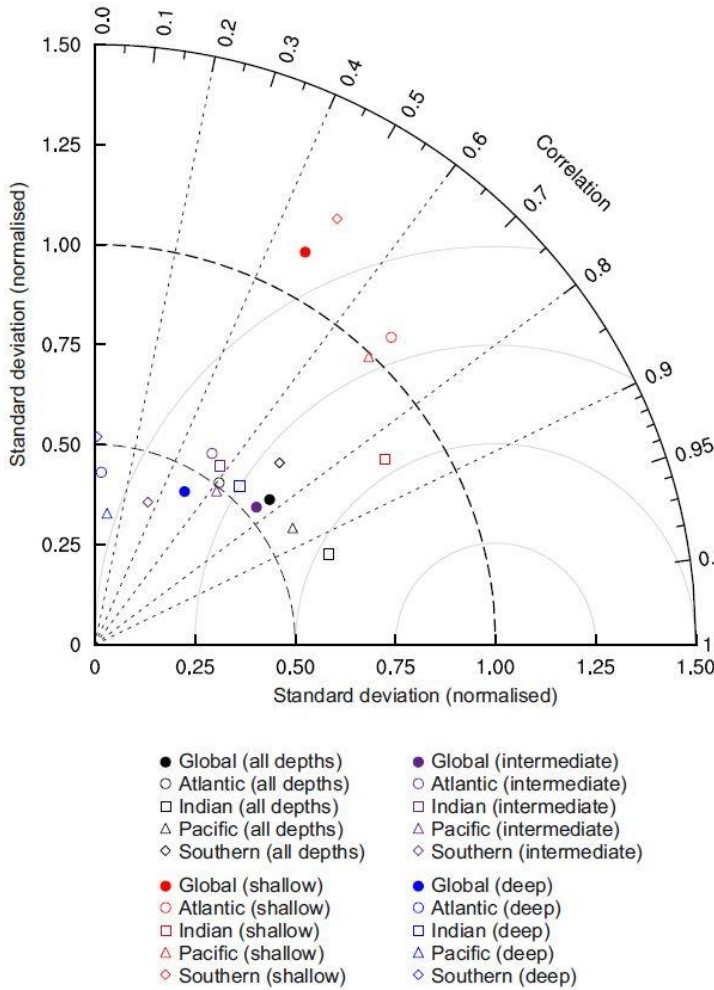

**Figure 15:** Taylor plot of $^{14}$C age relative to the idealised water age at the end of the spin-up simulation (years 9900 to 10,000). The data are separated by basin (shapes) and water depth (colours): shallow (0 to 550 m), intermediate (550 to 2500 m), and deep (2500 to 5500 m). A perfect simulation would have a correlation coefficient of 1 and a normalised standard deviation (simulated standard deviation/observed standard deviation) of 1.







**Figure 16:** Global and regional depth profiles of water age (orange), $^{14}$C age (solid black), and DIC concentration (dotted black) at the end of the spin-up simulation (years 9900 to 10,000). The water ages and $^{14}$C ages use the bottom axis, and the DIC concentrations use the top axis. The regions are outlined in Figure 7: Caribbean Sea (CS), Drake Passage (DP), eastern equatorial Atlantic upwelling zone (EEA_UP), eastern equatorial Pacific upwelling zone (EEP_UP), Labrador Sea (LS),
5   Northern Hemisphere Atlantic sub-tropical gyre (NH_ASG), Northern Hemisphere deep water formation region (NH_DWF), North Pacific (NP), Nova Scotia (NS), Southern Hemisphere Atlantic sub-tropical gyre (SH_ASG), Southern Hemisphere deep water formation region (SH_DWF), Southern Hemisphere Pacific sub-tropical gyre (SH_PSG), Southern Ocean upwelling zone (SO_UP), and Tasman Sea (TS).





## Appendix A: Virtual fluxes

The standard equation for calculating the virtual flux to account for the dilution or concentration effect of surface freshwater fluxes is:

$$\frac{d^{12}C}{dt} = {}^{12}C \cdot \frac{(E-P)}{dz} \tag{A1}$$

where $E$ is evaporation, $P$ is precipitation, and $dz$ is layer depth.

As we carry $^{14}C$ as a ratio ($^{14}C/^{12}C$), virtual fluxes are not required:

$$\frac{d\left(\frac{^{14}C}{^{12}C}\right)}{dt} = \frac{{}^{12}C \cdot \frac{d^{14}C}{dt} - {}^{14}C \cdot \frac{d^{12}C}{dt}}{\left({}^{12}C\right)^2} \tag{A2}$$

$$\frac{d\left(\frac{^{14}C}{^{12}C}\right)}{dt} = \frac{1}{^{12}C} \cdot \left[{}^{14}C \cdot \frac{(E-P)}{dz}\right] - \frac{^{14}C}{\left({}^{12}C\right)^2} \cdot \left[{}^{12}C \cdot \frac{(E-P)}{dz}\right] \tag{A3}$$

$$\frac{d\left(\frac{^{14}C}{^{12}C}\right)}{dt} = 0 \tag{A4}$$

## Appendix B: Air-sea gas exchange equations

### B.1 Abiotic $^{14}C$

The standard equation for calculating the change in abiotic $DI^{14}C$ due to air-sea gas exchange is:

$$\frac{d^{14}C}{dt} = PV \cdot \left(C_{sat} \cdot \frac{^{14}A}{^{12}A} - C_{surf} \cdot \frac{^{14}C}{^{12}C}\right) \tag{B1}$$

where $PV$ is the piston velocity (Eq. (4)), $C_{sat}$ is the saturation concentration of atmospheric $CO_2$ (in mol m⁻³), $C_{surf}$ is the
surface aqueous concentration of $CO_2$ (in mol m⁻³), and $^{14}A/^{12}A$ and $^{14}C/^{12}C$ are the $^{14}C/^{12}C$ ratios of the atmosphere and DIC, respectively.

The equation for calculating the change in abiotic $DI^{14}C/ DI^{12}C$ due to air-sea gas exchange is:

$$\frac{d\left(\frac{^{14}C}{^{12}C}\right)}{dt} = \frac{{}^{12}C \cdot \frac{d^{14}C}{dt} - {}^{14}C \cdot \frac{d^{12}C}{dt}}{\left({}^{12}C\right)^2} \tag{B2}$$

$$\frac{d\left(\frac{^{14}C}{^{12}C}\right)}{dt} = \frac{1}{^{12}C} \cdot \left[PV \cdot \left(C_{sat} \cdot \frac{^{14}A}{^{12}A} - C_{surf} \cdot \frac{^{14}C}{^{12}C}\right)\right] - \frac{^{14}C}{\left({}^{12}C\right)^2} \cdot \left[PV \cdot (C_{sat} - C_{surf})\right] \tag{B3}$$

$$\frac{d\left(\frac{^{14}C}{^{12}C}\right)}{dt} = \frac{1}{^{12}C} \cdot PV \cdot \left[\left(C_{sat} \cdot \frac{^{14}A}{^{12}A} - C_{sat} \cdot \frac{^{14}C}{^{12}C}\right) - \left(C_{surf} \cdot \frac{^{14}C}{^{12}C} - C_{surf} \cdot \frac{^{14}C}{^{12}C}\right)\right] \tag{B4}$$

$$\frac{d\left(\frac{^{14}C}{^{12}C}\right)}{dt} = \frac{1}{^{12}C} \cdot PV \cdot C_{sat} \cdot \left(\frac{^{14}A}{^{12}A} - \frac{^{14}C}{^{12}C}\right) \tag{B5}$$





## B.2 Biotic $^{14}$C

The standard equation for calculating the change in biotic DI$^{14}$C due to air-sea gas exchange is:

$$\frac{d^{14}C}{dt} = \alpha_k \cdot \alpha_{aq \leftarrow g} \cdot PV \cdot \left( C_{sat} \cdot \frac{^{14}A}{^{12}A} - \frac{C_{surf} \cdot \frac{^{14}C}{^{12}C}}{\alpha_{DIC \leftarrow g}} \right) \tag{B6}$$

where $\alpha_k$ is the constant kinetic fractionation factor (0.99919), $PV$ is the piston velocity (Eq. (4)), $C_{sat}$ is the saturation concentration of atmospheric $CO_2$ (in mol m$^{-3}$), $C_{surf}$ is the surface aqueous concentration of $CO_2$ (in mol m$^{-3}$), $^{14}A/^{12}A$ and $^{14}C/^{12}C$ are the $^{14}$C/$^{12}$C ratios of the atmosphere and DIC, respectively, $\alpha_{aq \leftarrow g}$ is the temperature-dependent fractionation during gas dissolution:

$$\alpha_{aq \leftarrow g} = 0.9986 - (4.9 \times 10^{-6}) \times SST \ , \tag{B7}$$

and $\alpha_{DIC \leftarrow g}$ is the temperature-dependent fractionation between aqueous $CO_2$ and DIC:

$$\alpha_{DIC \leftarrow g} = 1.01051 - (1.05 \times 10^{-4}) \times SST. \tag{B8}$$

For each process, the isotopic enrichment factor ($\varepsilon$, Eq. (7)) for $^{14}$C is twice that of $^{13}$C.

The equation for calculating the change in biotic DI$^{14}$C/ DI$^{12}$C due to air-sea gas exchange is:

$$\frac{d\left(\frac{^{14}C}{^{12}C}\right)}{dt} = \frac{^{12}C \cdot \frac{d^{14}C}{dt} - ^{14}C \cdot \frac{d^{12}C}{dt}}{\left(^{12}C\right)^2} \tag{B9}$$

$$\frac{d\left(\frac{^{14}C}{^{12}C}\right)}{dt} = \frac{1}{^{12}C} \cdot \left[ \alpha_k \cdot \alpha_{aq \leftarrow g} \cdot PV \cdot \left( C_{sat} \cdot \frac{^{14}A}{^{12}A} - \frac{C_{surf} \cdot \frac{^{14}C}{^{12}C}}{\alpha_{DIC \leftarrow g}} \right) \right] - \frac{^{14}C}{\left(^{12}C\right)^2} \cdot \left[ PV \cdot (C_{sat} - C_{surf}) \right] \tag{B10}$$

$$\frac{d\left(\frac{^{14}C}{^{12}C}\right)}{dt} = \frac{1}{^{12}C} \cdot PV \cdot \left[ \alpha_k \cdot \alpha_{aq \leftarrow g} \cdot \left( C_{sat} \cdot \frac{^{14}A}{^{12}A} - \frac{C_{surf} \cdot \frac{^{14}C}{^{12}C}}{\alpha_{DIC \leftarrow g}} \right) - \left( \frac{^{14}C}{^{12}C} \cdot \left[ C_{sat} - C_{surf} \right] \right) \right] \tag{B11}$$