# Peer review of "Simulating oceanic radiocarbon with the FAMOUS GCM: implications for its use as a proxy for ventilation and carbon uptake"

_Biogeosciences, 2019_

## Referee Comment (RC1) · Anonymous Referee #1 · 15 Nov 2019

Dentith et al. describe the implementation of radiocarbon (14C, or $\Delta$14C when considering the normalized and fractionation-corrected 14C/12C ratio) into the ocean component of the FAMOUS atmosphere-ocean GCM and present two 14C simulations, one spin-up simulation with constant pre-industrial boundary conditions plus a "historical" simulation forced with transient values of atmospheric $CO_2$ and $\Delta$14C for 1765–2000 CE. The simulation results are compared with water column measurements available for the 1950s and the 1990s, and with 14C records from bivalves and deep-water corals spanning the period 1880–2000 CE. The model results are at least qualitatively in line with observations. FAMOUS simulates radiocarbon in two ways. The "abiotic" approach only considers uptake, transport and radioactive decay of normalized 14C/12C.

The "biotic" approach also considers isotopic fractionation due to air-sea gas exchange and the isotopic imprint of the biological pump. Comparing both approaches, Dentith et al. find that the "biotic" approach results in slightly elevated 14C/12C ratios in the deep sea (by about 20‰ in terms of $\Delta$14C). This result corroborates early findings stating that biological effects on 14C/12C are much smaller (<10%) than the effects of transport and radioactive decay (Fiadeiro, 1982). In addition, Dentith et al. compare simulated 14C water ages with "true" water ages according to an idealized age tracer. It turns out that for large parts of the oceans, a purely kinematic interpretation of 14C/12C ages can lead to erroneous conclusions.

The paper is well written, the presentation of the results is mostly clear, and the literature is comprehensive. It may be published if the following issues are amended:

Major issue:

As described in Section 2.3.2, the model setup does not allow for meridional variability of atmospheric $\Delta$14C. Therefore, the historical simulation misses the strong interhemispheric $\Delta$14C gradient in the atmosphere of up to 400‰ (e.g., Fig. 1 in the manuscript) due to nuclear weapons testing. As a consequence, the model may underestimate the actual spatiotemporal 14C gradients in the ocean since the late 1950s. This shortcoming may distort the evolution of bomb 14C transients discussed in Section 3.3. It may also lead to biased post-bomb 14C distributions in subsurface and deeper waters which are discussed in Section 3.2. I would strongly recommend repeating the historical simulation forced with hemispheric averages of atmospheric $\Delta$14C. The implementation should not be too difficult (the model has to read in three files provided by OCMIP-2).

Further issues and comments (P = page, L = line, SM = supplementary material):

P 3, L 6: The value of the half-life should be consistent with the updated value of 5700 years promoted in the SM.

P 3, L 28: "Indian" should read "Indien"

P 4, L 28: The models MOM (Toggweiler et al., 1989), LOCH (Mouchet, 2013) and LOVEVLIM (e.g., Menviel et al. 2017) could be added to the list.

P 7, L 28: (660 / Sc)**-0.5 should read (Sc / 660)**-0.5 (e.g., Orr et al. 2017, equation (12))

P 7, L 31: Missing reference (I guess it is Wanninkhof 1992)

P 8, L 8 and P 22, L 27: I could not find the paper at GMDD. What is the current status of the manuscript?

P 8, L 12: 14 and 13 should be subscripts

P 8, L 22: "$\delta$13C is close to zero": this is the only case for DIC but not for phytoplankton

P 9, L 10: This is not correct, Stuiver and Pollach (1977) employ 5568 years.

P 9, L 16: What about turbulent diffusion, where is it included?

P 9, L 26–28 and Figure S1: The OCMIP-2 steady state criterion is somewhat different, demanding that 98% of the ocean volume has a $\Delta$14C drift of less than 0.001‰ per year (see also Aumont et al. 1998). The total 14C inventory may have stabilized while there might be still an ongoing internal redistribution of 14C. Have you checked this?

P 13, L 5–6: "(...) the simulated $\Delta$14C gradient between the surface ocean and approximately 1000 m depth is shallower than observed (. . .). This is visible in all regions, except the Northern Hemisphere deep water formation region (NH_DWF) and the North Pacific (NP) (. . .)." According to Figure 8 the opposite is true (indicating steeper simulated gradients everywhere except for NH_DW and NP).

P 13, L 23: "(. . .) due to the sparsity of data." You might wish to compare your results with post-bomb $\Delta$14C (bottle) data provided by GLODAPv2 (https://odv.awi.de/data/ocean/ glodap-v22019-bottle-data/).

P 17, L 19: This is an interesting result supporting and quantifying the notion by Fiadeiro (1982) that 14C/12C can roughly be regarded as radio-conservative tracer in the deep sea.

P 18, L 1–6: The simulated 14C ages reflect local ageing, transport, biogeochemistry, and radioactive decay. On the other hand, the simulated ideal water ages only reflect local ageing and transport. As there is no radioactive decay of the age tracer, the age differences cannot be entirely explained with biogeochemical effects.

P 18, L 15–16: "(...) the water ages generally increase with depth because they are a simple function of advection." I disagree, there may be considerable mixing leading to nonlinear ageing of water parcels. This is seen in many tracers records.

P 31, Figure 4 (b): I would prefer to see the differences between simulated and observed values, analogously to Figure 2.

Figures 8, 10, 11, 16, S4, S5, and S6: Lines and dots are blurred.

P 38, Figure 11: The line plots should be replaced with Hovmöller diagrams.

SM, P 2: "OCMIP" should read "OCMIP-2"

References:

Aumont, O.; Orr, J. C.; Jamous, D.; Monfray, P.; Marti, O.; Madec, G. A Degradation Approach to Accelerate Simulations to Steady-State in a 3-D Tracer Transport Model of the Global Ocean. Climate Dynamics 1998, 14 (2), 101–116.

Fiadeiro, M. E. Three-Dimensional Modeling of Tracers in the Deep Pacific Ocean, II. Radiocarbon and the Circulation. Journal of Marine Research 1982, 40, 537–550.

Menviel, L.; Yu, J.; Joos, F.; Mouchet, A.; Meissner, K. J.; England, M. H. Poorly Ventilated Deep Ocean at the Last Glacial Maximum Inferred from Carbon Isotopes: A Data-Model Comparison Study. Paleoceanography 2017, 32 (1), 2016PA003024.

Mouchet, A. The Ocean Bomb Radiocarbon Inventory Revisited. Radiocarbon 2013,

55, 1580–1594. Orr, J. C.; Najjar, R. G.; Aumont, O.; Bopp, L.; Bullister, J. L.; Danabasoglu, G.; Doney, S. C.; Dunne, J. P.; Dutay, J.-C.; Graven, H.; et al. Biogeochemical Protocols and Diagnostics for the CMIP6 Ocean Model Intercomparison Project (OMIP). Geosci. Model Dev. 2017, 10 (6), 2169–2199.

Orr, J. C.; Najjar, R. G.; Aumont, O.; Bopp, L.; Bullister, J. L.; Danabasoglu, G.; Doney, S. C.; Dunne, J. P.; Dutay, J.-C.; Graven, H.; et al. Biogeochemical Protocols and Diagnostics for the CMIP6 Ocean Model Intercomparison Project (OMIP). Geosci. Model Dev. 2017, 10 (6), 2169–2199.

---

## Referee Comment (RC2) · Anonymous Referee #2 · 30 Nov 2019

Review of 'Simulating oceanic radiocarbon with the FAMOUS GCM: implications for its use as a proxy for ventilation and carbon uptake' by Dentith et al., submitted to Biogeosciences (manuscript bg-2019-365).

In this work Dentith et al. describe the implementation of radiocarbon and age tracers in the ocean component of the FAMOUS model. Model performances are evaluated against data for the pre- and post- bomb periods. They then assess the role of biological processes in driving ocean radiocarbon distributions. Eventually, an analysis of the departures between radiocarbon ages and water ages is provided.

The paper is very well written and structured. I also appreciate the throughout model assessment and careful comparison with archives and data.

However I have several concerns with respect to the interpretation of the $\delta^{14}C$ and age distributions. In addition, the way radiocarbon is represented in the model calls for a strong assumption on air-sea $CO_2$ equilibrium state. This is why I do not recommend immediate publication of this paper.

**Main comments**

**1) Modeling of radiocarbon**

- The method presented in Section 2.2.2. is not that described in the OCMIP-2 protocol. In OCMIP-2, $^{14}C$ and DIC were both prognostic variables constrained by adequate boundary conditions at the air-sea interface (Orr et al., 1999); $\Delta^{14}C$ was then obtained by computing the ratio of these two quantities.

  The method in the present manuscript (modeling of the $^{14}C/C$ ratio) is that first suggested by Fiadeiro (1982) and popularized by Toggweiler et al. (1989). The only difference is that here the DIC value used for scaling the air-sea flux of the ratio (Eq. 6) is not constant, similar to what is done in Butzin et al. (2017); the impact of such a change is expected to be minimal.

  Though ideal for assessing the ocean ventilation (Broecker et al., 1961; Maier-Reimer, 1993) this method is not fit for addressing bomb radiocarbon at a time of major change in atmospheric $CO_2$ since it implicitly assumes that local air-sea $CO_2$ disequilibrium remains constant with time (Mouchet, 2013); this significantly affects the $^{14}C$ invasion rate into the ocean.

- I seriously wonder why not represent in the model the individual carbon species $^{13}C$ and $^{14}C$ rather than their ratios. It would not call for additional tracers. Proceeding so would also guarantee that all important processes and timescales are considered; this is especially important when addressing the anthropogenic era during which rapid and significant changes occur in all three carbon species.

  An additional advantage would be that all fluxes are more straightforward to implement in the model reducing so the risk of mistakes.

**2) Interpretation of radiocarbon anomalies**

- There is some confusion among $\Delta^{14}C$ and $\delta^{14}C$ in section 2.2.4 (page 9, lines 1 to 5). $\Delta^{14}C$ is the normalized $^{14}C/C$ ratio corrected for isotopic fractionation; that is $\Delta^{14}C$ reflects the $^{14}C/C$ ratio which would be observed if there was no fractionation during any of the processes involved in the building of the material under study.

  In the 'abiotic' framework one hypothesizes that fractionation is negligible – this assumption applies to all fractionation processes; or in short there is no fractionation whatever the process, and $\Delta^{14}C = \delta^{14}C$. Hence the $\Delta^{14}C$ values predicted by the 'abiotic' model may be directly compared to measured $\Delta^{14}C$ values, as has been done by many (Toggweiler et al. 1989; De Vries and Primeau, 2011; Mouchet, 2013; Butzin et al., 2017).

  The 'biotic' $\delta^{14}C$ must be corrected for fractionation as in authors' Eq. (8) to be compared to observed $\Delta^{14}C$. In the end the 'biotic' $\Delta^{14}C$ and the 'abiotic' $\Delta^{14}C$ should be very close (Bacastow and Maier-Reimer, 1990).

  In contrast, $\delta^{14}C$ or the inventory (not the normalized ratio) of $^{14}C$ atoms is lower by about 5% when neglecting fractionation; this aspect is thoroughly discussed in Orr et al. (2017).

- There is further confusion in the definition of 'Abiotic' and 'Biotic' processes (sections 2.2.2 and 2.2.3). Fractionation during air-sea exchange is not a biological process. Impact from the biology is twofold: 1) fractionation during soft tissue production, with a preference for the lighter isotopes, and 2) modification of the air-sea $CO_2$ gradient pattern.

- The dominant control on the differences between $\delta^{14}C$-biotic and $\delta^{14}C$-abiotic values (section 3.4) is fractionation during the air-sea transfer, not the biological pump.

  Indeed, according to the authors' equations (B7) and (B8), air-sea fractionation at equilibrium results in an ocean $^{14}C$ enrichment of 21 ‰ at 0°c to 15.8 ‰ at 25°C (consider the first part of Eq. (B11) while assuming equilibrium). Biological activity would increase by only 4 ‰ the surface values and slightly decrease the ratios at depth. However, globally the difference between $\delta^{14}C$-biotic and $\delta^{14}C$-abiotic is nearly completely due to fractionation during air-sea processes.

- It would be of most interest to evaluate the departures between $\Delta^{14}C$ with and without biology, as well as departures between the amount of bomb radiocarbon in the ocean obtained by each method.

**3) Water age**

I must acknowledge that I am not happy at all with the water age results nor with the interpretation of the differences between radiocarbon ages and water ages. Something seems wrong in the implementation of the water age. The discussion of the factors controlling differences between radiocarbon-based and water ages is not based on any evidence.

- It is striking that modeled water ages are everywhere much larger than the radiocarbon age in the deep ocean (Figs 14 and 16) at the exception of polar areas. Given the non-zero preformed radiocarbon ages (reservoir ages) one would expect the opposite relationship; i.e., that water ages are smaller than radiocarbon ages in the deep (e.g., Campin et al., 1999;

Franke et al., 2008; Khatiwala et al. 2012; Koeve et al. 2015). How exactly is the water age computed in the model? Which processes do control its distribution?

- On page 18, lines 15-16 it is mentioned that "… water ages ... are a simple function of advection.". Should we interpret that water age does not experience mixing or diffusion? This would be erroneous.

- In the discussion of regional distributions (page 18, lines 18-25) how do you assess the relative roles of circulation and solubility in controlling the difference between water age and radiocarbon age? There are no solid arguments allowing to conclude in the domination of the one or the other in any region. This paragraph is not based on any firm evidence.

- Similar wishful thinking occurs on lines 28-32 on the same page. The solubility (surface temperature) has nothing to do with water properties at depth in the Indian Ocean.

- Differences between water age and radiocarbon age have already been addressed in numerous academic works; e.g., Campin et al. (1999), Delhez et al. (2003), Gebbie and Huybers (2012), Koeve et al. (2015). This is a non-exhaustive list that I recommend as a start.

**Miscellaneous**

Abstract, line 13 : What do you mean by 'over-deep' NADW?

Page 5, line 9: "*...post-bomb deep ocean $\Delta^{14}C$ (i.e. natural $^{14}C$ distributions)*" I would not qualify the post-bomb deep radiocarbon as 'natural'; while some large areas might still hold the pristine signal many other deep ocean areas with younger age could be contaminated by the bomb signal.

Pages 7 and 8, lines 14-18 and 24-26: these lines are unnecessary since the 'Abiotic $^{14}C/^{12}C$' flux (Eq. 6) does not call for the computation of aqueous $CO_2$.

Page 7, equation 2: is the constant 100 or 1000?

Page 7, lines 17-18: "*In the calculation of aqueous CO2, we use the carbonic acid constants of Roy et al. (1993) as opposed to Millero (1995) because this is consistent with the formulation of CO2 solubility used in other areas of the model.*" What is the rationale for requesting coherency between solubility and carbonate dissociation constants? These are two totally different topics. Shouldn't the dissociation constants be consistent among each others and with the pH scale used in the model?

Page 9, subsection title and line 16 (L term): does the transport only include advection in the model?

Section 3.3: why restrict the transient study to the North Atlantic? There are existing coral records in various other locations too (e.g. Druffel, 2002).

Page 16, lines 31-32: "*We propose that this asymmetry relates to the age of the waters that are being upwelled in each basin.*" The 'we propose' formulation is confusing with respect to the correct explanation that follows.

Page 19, lines 5-6: "*However, Campin et al. (1999) did not account for isotopic fractionation in their study, nor was their $^{14}C$ tracer cycled through the marine biological pump.*" This sentence is

out of topic. Campin et al. (1999) represented the normalized $\Delta^{14}C$ ratio in their model. Fractionation effects are therefore canceled and would not explain any difference in age behavior. Campin et al. (1999) did indeed not consider biological activity, but its impact is rather small on $\Delta^{14}C$ contours (Bacastow and Maier-Reimer, 1990). Ages are computed with the help of $\Delta^{14}C$ not with the $\delta^{14}C$ ratios.

Article by Dentith et al. (submitted), quoted at different places in the text, is nowhere to be found on the editor (gmd) site. It would have been an advantage to have been able to consult it in order of understanding the methodology briefly presented in section 2.2.3.

**References**

Broecker WS, Gerard RD, Ewing M, Heezen BC. Geochemistry and physics of ocean circulation. In: Sears M, editor. Oceanography. Washington: AAAS. Publication 67, p 301-322, 1961.

Butzin, M., P. Köhler, and G. Lohmann, Marine radiocarbon reservoir age simulations for the past 50,000 years, Geophys. Res. Lett. 44, 8473–8480, doi:10.1002/2017GL074688, 2017.

Campin, J., Fichefet, T., Duplessy, J. Problems with using radiocarbon to infer ocean ventilation rates for past and present climates. Earth Planet. Sci. Lett. 165, 17–24, 1999.

Delhez E.J.M., E. Deleersnijder, A. Mouchet and J-M. Beckers. A note on the age of radioactive tracers, Journal of Marine Systems, 38, 277-286, 2003.

Druffel, E. R. M. Radiocarbon in corals: records of the carbon cycle, surface circulation and climate. Oceanography,15, 122–127. 2002.

Fiadeiro, M. E. Three-Dimensional Modeling of Tracers in the Deep Pacific Ocean, II. Radiocarbon and the Circulation. Journal of Marine Research, 40, 537–550, 1982.

Franke, J., M. Schulz, A. Paul, and J. F. Adkins. Assessing the ability of the 14C projection-age method to constrain the circulation of the past in a 3-D ocean model, Geochem. Geophys. Geosyst., 9, Q08003, doi:10.1029/2008GC001943, 2008.

Gebbie, G., and P. Huybers. The mean age of ocean waters inferred from radiocarbon observations: sensitivity to surface sources and accounting for mixing histories. J. Phys. Oceanogr., 42, doi: 10.1175/JPO-D-11-043.1, pp. 291–305, 2012.

Khatiwala, S., F. Primeau, and M. Holzer. Ventilation of the deep ocean constrained with tracer observations and implications for radiocarbon estimates of ideal mean age. Earth and Planet. Sci. Lett., doi:10.1016/j.epsl.2012.01.038, 2012.

Koeve, W., Wagner, H., Kähler, P., and Oschlies, A.: 14C-age tracers in global ocean circulation models, Geosci. Model Dev., 8, 2079–2094, https://doi.org/10.5194/gmd-8-2079-2015, 2015.

Maier-Reimer E. Geochemical cycles in an ocean general circulation model. Preindustrial tracer distributions. Global Biogeochemical Cycles 7: 645-677, 1993.

Mouchet A. The Ocean Bomb Radiocarbon Inventory Revisited. Radiocarbon 55: 1580-1594, 2013.

Orr, J. C., Najjar, R. G., Aumont, O., Bopp, L., Bullister, J. L., Danabasoglu, G., Doney, S. C., Dunne, J. P., Dutay, J.-C., Graven, H., Griffies, S. M., John, J. G., Joos, F., Levin, I., Lindsay, K., Matear, R. J., McKinley, G. A., Mouchet, A., Oschlies, A., Romanou, A., Schlitzer, R., Tagliabue, A., Tanhua, T., and Yool, A.: Biogeochemical protocols and diagnostics for the CMIP6 Ocean Model Intercomparison Project (OMIP), Geosci. Model Dev., 10, 2169–2199, https://doi.org/10.5194/gmd-10-2169-2017, 2017.

Orr, J. C., Najjar, R., Sabine, C. L., and Joos, F.: Abiotic-HOWTO, Internal OCMIP Report, LSCE/CEA Saclay, Gif-sur-Yvette, France, 25 pp., http://ocmip5.ipsl.jussieu.fr/OCMIP/phase2/simulations/Abiotic/HOWTO-Abiotic.html, 1999.

Toggweiler JR, Dixon K, Bryan K. Simulations of radiocarbon in a coarse-resolution world ocean model. 1. Steady state prebomb distributions. Journal of Geophysical Research 94:8217–42. 1989.

---

## Author Comment (AC2) · 24 Feb 2020

**Response to reviewer #2**

In this work Dentith et al. describe the implementation of radiocarbon and age tracers in the ocean component of the FAMOUS model. Model performances are evaluated against data for the pre- and post- bomb periods. They then assess the role of biological processes in driving ocean radiocarbon distributions. Eventually, an analysis of the departures between radiocarbon ages and water ages is provided.

The paper is very well written and structured. I also appreciate the throughout model assessment and careful comparison with archives and data.

However I have several concerns with respect to the interpretation of the $\delta^{14}C$ and age distributions. In addition, the way radiocarbon is represented in the model calls for a strong assumption on air-sea $CO_2$ equilibrium state. This is why I do not recommend immediate publication of this paper.

*We would like to thank reviewer #2 for their feedback on our manuscript. We have considered all of the comments carefully and addressed them in turn below, with our responses in blue-italics.*

**Main comments**

*1) Modeling of radiocarbon*

- The method presented in Section 2.2.2. is not that described in the OCMIP-2 protocol. In OCMIP-2, $^{14}C$ and DIC were both prognostic variables constrained by adequate boundary conditions at the air-sea interface (Orr et al., 1999); $\Delta^{14}C$ was then obtained by computing the ratio of these two quantities.

  The method in the present manuscript (modeling of the 14C/C ratio) is that first suggested by Fiadeiro (1982) and popularized by Toggweiler et al. (1989). The only difference is that here the DIC value used for scaling the air-sea flux of the ratio (Eq. 6) is not constant, similar to what is done in Butzin et al. (2017); the impact of such a change is expected to be minimal.

  Though ideal for assessing the ocean ventilation (Broecker et al., 1961; Maier-Reimer, 1993) this method is not fit for addressing bomb radiocarbon at a time of major change in atmospheric $CO_2$ since it implicitly assumes that local air-sea $CO_2$ disequilibrium remains constant with time (Mouchet, 2013); this significantly affects the $^{14}C$ invasion rate into the ocean.

*We will revise the manuscript to clarify that, in our model, radiocarbon is transported and exchanged with the atmosphere as $DI^{14}C/DI^{12}C$, rather than as individual C isotopes as in the OCMIP-2 protocol. It is only in the biological part of the model that we exchange $^{14}C$ (rather than the $DI^{14}C/DI^{12}C$ ratio) through the different biological carbon pools (see our answer to the next comment). However, our implementation is different to that tested in Mouchet (2013), who assumed constant and uniform DIC concentrations when they implement radiocarbon as a ratio. As pointed out by the reviewer, DIC is freely evolving spatially and temporally in our model and air-sea gas exchange of carbon is dependent on this variable DIC concentration. It is therefore not the case "that local air-sea $CO_2$ disequilibrium remains constant with time" as in Mouchet (2013). Our implementation of air-sea gas exchange of radiocarbon as a ratio is mathematically equivalent to the OCMIP-2 protocol. The formulation of air sea gas exchange of the ratio ($DI^{14}C/DI^{12}C$) is not explicitly dependent on local air-sea $CO_2$ disequilibrium (Eq. (6) in the original manuscript). However, the air-sea gas exchange of $^{12}C$ is dependent on local air-sea $CO_2$ disequilibrium (Eq. (3) in the original manuscript), and therefore the changes of $DI^{14}C$ as a consequence of air-sea gas exchange are dependent on local air-sea $CO_2$ disequilibrium, which varies through time. We do agree that due to numerical discretisation schemes, carrying radiocarbon as a ratio is not equivalent to carrying both $^{12}C$ and $^{14}C$ separately (Mouchet,*

*2013). For example, our implementation removes the need for virtual flux corrections of $^{14}C$ as a consequence of evaporation and precipitation as is otherwise required in the OCMIP protocol. As a result, our ratio implementation may not be the most accurate radiocarbon representation for evaluating air-sea gas exchange during the historical period, but FAMOUS would not be the right tool for this anyway and this is not the purpose of our study. The indended use of our implementation is to study millenial changes in ocean circulation in the geological past. The comparison to the bomb inventories in this manuscript is intended as a verification of our model and not as a means to determine present-day air-sea gas exchange. This will be clarified in the revised version of our manuscript along with the addition of a discussion on the implications of carrying radiocarbon as a ratio.*

- I seriously wonder why not represent in the model the individual carbon species $^{13}C$ and $^{14}C$ rather than their ratios. It would not call for additional tracers. Proceeding so would also guarantee that all important processes and timescales are considered; this is especially important when addressing the anthropogenic era during which rapid and significant changes occur in all three carbon species.

  An additional advantage would be that all fluxes are more straightforward to implement in the model reducing so the risk of mistakes.

*The abiotic $^{14}C$ legacy code used a ratio in model units and we implemented the biotic $^{14}C$ in the same manner for consistency. We thought about this point very carefully before undertaking the implementation, and we ultimately decided on the presented approach due to two major advantages:*
1. *the use of a ratio negates the need for virtual fluxes;*
2. *model units reduce errors and numerical instabilities associated with small numbers.*

*However, the isotopes are carried as individual species within the biological scheme where the code is more complex. Specifically, for consistency with the standard biological tracers and to reduce the risk of mistakes, the $^{14}C$ contents of phytoplankton ($^{14}P$), zooplankton ($^{14}Z$), and detritus ($^{14}D$) are expressed in mmol-N m$^{-3}$. The DI$^{14}C$/DI$^{12}C$ values are converted from a ratio in model units to normalised DI$^{14}C$ concentrations before entering the biological pump. The conversion is reversed at the end of each timestep. This is outlined in the figure below, which is from our GMD manuscript (Dentith et al., submitted: https://www.geosci-model-dev-discuss.net/gmd-2019-250/0 - we appreciate that a technical hold up meant this figure was not available to the reviewer until recently).*

[Figure]

**Figure R1:** *Blue boxes represent permanent carbon pools. Grey boxes represent temporary carbon pools. The orange box represents the prescribed atmospheric carbon pool. The dashed line represents fluxes of $^{14}C/^{12}C$. Solid lines represent fluxes of $^{14}C$. Dot-dashed lines represent processes that occur below the lysocline (approximately 2500 m below sea level). The dotted line represents the reflux of detrital material from the seafloor to the surface layer. Red lines represent fractionation effects. In this study, we ran without fractionation during calcium carbonate formation ($\alpha_{CaCO3} = 1.0$).*

*2) Interpretation of radiocarbon anomalies*

- There is some confusion among $\Delta^{14}C$ and $\delta^{14}C$ in section 2.2.4 (page 9, lines 1 to 5). $\Delta^{14}C$ is the normalized $^{14}C/C$ ratio corrected for isotopic fractionation; that is $\Delta^{14}C$ reflects the $^{14}C/C$ ratio which would be observed if there was no fractionation during any of the processes involved in the building of the material under study.

In the 'abiotic' framework one hypothesizes that fractionation is negligible – this assumption applies to all fractionation processes; or in short there is no fractionation whatever the process, and $\Delta^{14}C = \delta^{14}C$. Hence the $\Delta^{14}C$ values predicted by the 'abiotic' model may be directly compared to measured $\Delta^{14}C$ values, as has been done by many (Toggweiler et al. 1989; De Vries and Primeau, 2011; Mouchet, 2013; Butzin et al., 2017).

The 'biotic' $\delta^{14}C$ must be corrected for fractionation as in authors' Eq. (8) to be compared to observed $\Delta^{14}C$. In the end the 'biotic' $\Delta^{14}C$ and the 'abiotic' $\Delta^{14}C$ should be very close (Bacastow and Maier-Reimer, 1990).

In contrast, $\delta^{14}C$ or the inventory (not the normalized ratio) of $^{14}C$ atoms is lower by about 5% when neglecting fractionation; this aspect is thoroughly discussed in Orr et al. (2017).

- There is further confusion in the definition of 'Abiotic' and 'Biotic' processes (sections 2.2.2 and 2.2.3). Fractionation during air-sea exchange is not a biological process. Impact from the biology is twofold: 1) fractionation during soft tissue production, with a preference for the lighter isotopes, and 2) modification of the air-sea $CO_2$ gradient pattern.

- The dominant control on the differences between $\delta^{14}C$-biotic and $\delta^{14}C$-abiotic values (section 3.4) is fractionation during the air-sea transfer, not the biological pump.

  Indeed, according to the authors' equations (B7) and (B8), air-sea fractionation at equilibrium results in an ocean $^{14}C$ enrichment of 21 ‰ at 0°c to 15.8 ‰ at 25°C (consider the first part of Eq. (B11) while assuming equilibrium). Biological activity would increase by only 4 ‰ the surface values and slightly decrease the ratios at depth. However, globally the difference between $\delta^{14}C$-biotic and $\delta^{14}C$-abiotic is nearly completely due to fractionation during air-sea processes.

- It would be of most interest to evaluate the departures between $\Delta^{14}C$ with and without biology, as well as departures between the amount of bomb radiocarbon in the ocean obtained by each method.

*In response to the four comments above, the rationale behind our original analysis was as follows:*

*Originally, we compared the $\Delta^{14}C_{abiotic}$ to the $\Delta^{14}C_{biotic}$, where $\Delta^{14}C_{abiotic} = \delta^{14}C_{abiotic}$. However, because $\delta^{13}C_{DIC}$ is close to zero, the dominant component of the isotopic fractionation correction is the normalisation relative to the mean value of terrestrial wood (25 ‰), which results in the $\Delta^{14}C_{biotic}$ values being approximately 50 ‰ lower than the $\delta^{14}C_{biotic}$ values. As outlined in Section 3.4 of our manuscript, the $\delta^{14}C_{biotic}$ values are approximately 20 ‰ higher than the $\delta^{14}C_{abiotic}$ values because the biotic tracer accounts for the preferential uptake of $^{12}C$ during primary productivity, which leaves the DIC pool enriched in $^{14}C$, whereas the abiotic tracer does not account for this effect. Applying the isotopic fractionation correction, the $\Delta^{14}C_{biotic}$ values therefore end up being approximately 30 ‰ lower than the $\Delta^{14}C_{abiotic}$ values (i.e. comparing $\Delta^{14}C_{abiotic}$ to $\Delta^{14}C_{biotic}$ artifically reverses the relationship between the two tracers).*

*When we showed these results to analytical biogeochemists (comparing $\Delta^{14}C_{abiotic}$ to $\Delta^{14}C_{biotic}$), they found them to be conceptually confusing because, as outlined in our manuscript, the biotic values are expected to be higher than the abiotic values due to the aforementioned preferential removal of $^{12}C$ during photosynthesis, the effect of which is included in the biotic tracer, but not in the abiotic tracer. Based on their input, we think that the best way to communicate the difference between the abiotic and the biotic tracers is to present the comparison between uncorrected $\delta^{14}C_{biotic}$ and $\delta^{14}C_{abiotic}$.*

*We acknowledge that this approach is different to previous studies, such as Toggweiler et al. (1989) and Jahn et al. (2015), who present their abiotic tracers as $\Delta^{14}C$. However, in reality, the abiotic tracer is unrealistic because it neglects important processes such as isotopic fractionation and the cycling of $^{14}C$ through the marine carbon cycle. In the absence of a biotic implementation, we concur that it is*

*valid to compare an abiotic tracer to observations as $\Delta^{14}C$ because abiotic $\Delta^{14}C$ is a useful first-order representation of the processes that are important for the distribution of oceanic $\Delta^{14}C$ (air-sea gas exchange, advection, and radioactive decay). When both an abiotic and a biotic formulation are included in the same model, however, the usefulness of comparing the two tracers is to examine the differences between the simulated fields to improve our understanding of the processes that are important for the distribution of oceanic $^{14}C$. Consequently, we only compare $\Delta^{14}C_{biotic}$ to observations (because this is a more complete representation of reality than $\Delta^{14}C_{abiotic}$), and we compare the biotic and abiotic tracers as uncorrected $\delta^{14}C_{biotic}$ and $\delta^{14}C_{abiotic}$.*

*We have discussed three potential approaches to this comparison with analytical biogeochemists (Figure R2):*

- *Uncorrected $\delta^{14}C$ (biotic and abiotic), as presented in our original manuscript*
- *Corrected $\Delta^{14}C$ (biotic and abiotic). This comparison is almost identical to the uncorrected difference because the $2(\delta^{13}C + 25)$ part of the correction is much larger than the $(1 + \delta^{14}C/1000)$ part.*
- *Uncorrected abiotic ($\Delta^{14}C=\delta^{14}C$) and corrected biotic $\Delta^{14}C$, as per Jahn et al. (2015) and the preferred approach of the reviewer.*

*However, our preferred response to the four comments above would be to remove the abiotic-biotic comparison from our revised manuscript, because we have not yet found a way to present these results to both biogeochemical modellers and analytical biogeochemists without causing contention and confusion along the lines discussed above. We will therefore remove all mention of the 'abiotic' tracer implementation and results from the revised manuscript.*

[Figure]

***Figure R2:*** *Differences in the biotic and abiotic tracers in the surface ocean at the end of the spin-up simulation (years 9900 to 10,000, left) and during the 1990s (right): (a, b) uncorrected biotic δ¹⁴C minus uncorrected abiotic δ¹⁴C (as presented in Figure 12 of the original manuscript), (c, d) corrected biotic Δ¹⁴C minus corrected abiotic Δ¹⁴C, and (e, f) corrected Δ¹⁴C minus uncorrected abiotic δ¹⁴C, which is equivalent to Δ¹⁴C in other modelling studies (e.g. Toggweiler et al., 1989; Jahn et al., 2015). Note that in (c) and (d), the isotopic fractionation correction (Eq. 8 in the original manuscript) has been applied to the abiotic tracer (even though the abiotic tracer is not affected by isotopic fractionation effects) to place both tracers in the same reference frame (Δ¹⁴C) as observational studies. Also, note that in (e) and (f), the Δ¹⁴C anomalies are negative because the isotopic fractionation correction that is applied to the biotic tracer (≈50 ‰) is larger than the uncorrected difference between the two tracers (≈20 ‰). In all six sub-plots, darker colours correspond to the largest absolute anomalies.*

*3) Water age*

I must acknowledge that I am not happy at all with the water age results nor with the interpretation of the differences between radiocarbon ages and water ages. Something seems wrong in the implementation of the water age. The discussion of the factors controlling differences between radiocarbon-based and water ages is not based on any evidence.

- It is striking that modeled water ages are everywhere much larger than the radiocarbon age in the deep ocean (Figs 14 and 16) at the exception of polar areas. Given the non-zero preformed radiocarbon ages (reservoir ages) one would expect the opposite relationship; i.e., that water ages are smaller than radiocarbon ages in the deep (e.g., Campin et al., 1999; Franke et al., 2008; Khatiwala et al. 2012; Koeve et al. 2015). How exactly is the water age computed in the model? Which processes do control its distribution?

*The water age implementation is very simple and follows that by England (1995). The water age tracer is carried by the same tracer transport scheme as temperature, salinity and all other ocean tracers (nutrients, etc.), including our new $^{14}C$ tracer. This scheme computes the effects of diffusion, mixing and advection (using flux-limited Quadratic Upstream Interpolation for Convective Kinematics, QUICK).*

*One suggestion as to why our simulated water ages are older than the simulated $^{14}C$ ages at depth is that $^{14}C$ ages are not linear functions of $^{14}C$ concentrations, therefore they are not conserved during mixing (Gebbie and Huybers, 2012).*

- On page 18, lines 15-16 it is mentioned that "… water ages ... are a simple function of advection.". Should we interpret that water age does not experience mixing or diffusion? This would be erroneous.

*As above, the water age tracer is affected by advection, mixing, and diffusion. We will add these important details to the revised manuscript.*

- In the discussion of regional distributions (page 18, lines 18-25) how do you assess the relative roles of circulation and solubility in controlling the difference between water age and radiocarbon age? There are no solid arguments allowing to conclude in the domination of the one or the other in any region. This paragraph is not based on any firm evidence.

*We have assumed that the water ages are only affected by circulation (which we have called the physical component of the solubility pump) and that the $^{14}C$ ages are affected by both components of the solubility pump as well as the biological pump. However, in the abiotic-biotic comparison, we ascertained that the biological pump has a near-constant influence with depth, so could be corrected for. Following this logic, where the water ages and $^{14}C$ ages are similar, we infer that circulation is the dominant component of the solubility pump, and where they are not, we infer that the chemical component is more dominant.*

- Similar wishful thinking occurs on lines 28-32 on the same page. The solubility (surface temperature) has nothing to do with water properties at depth in the Indian Ocean.

*We will remove the following sentence from the revised manuscript: "Secondly, the average surface ocean temperature is between 4.5 °C and 18 °C higher than in the other basins, which means that the chemical component of the solubility pump is weaker."*

- Differences between water age and radiocarbon age have already been addressed in numerous academic works; e.g., Campin et al. (1999), Delhez et al. (2003), Gebbie and Huybers (2012), Koeve et al. (2015). This is a non-exhaustive list that I recommend as a start.

*Thank you for drawing our attention to these papers. Some of the requested information has already been synthesised by Dentith (2019) - http://etheses.whiterose.ac.uk/25427/. We will add a summary of this existing discussion and additional key points from a selection of the suggested papers to Section 3.5 of the revised manuscript. In particular, we note:*

- *Campin et al. (1999) implemented two passive tracers ($\Delta^{14}C$ and water age) in a 3° x 3° ocean-only model to examine the relative contributions of circulation and air-sea gas exchange to large-scale $^{14}C$ distributions, and thus the extent to which $^{14}C$ can be interpreted as a ventilation tracer. The authors identified a decoupling between the $^{14}C$ age and the actual age of water in the deep water formation regions. Specifically, the North Atlantic Deep Water (NADW) and Antarctic Bottom Water (AABW) had similar absolute ages, but the $^{14}C$ age of the AABW was systematically older than its water age and also the $^{14}C$ age of the NADW. In the Southern Ocean, the waters southwards of 60° S were isolated from the well ventilated ($^{14}C$-rich) sub-tropical surface waters, surface water residence times were short compared to the timescale for isotopic $^{14}CO_2$ equilibration, and compacted sea ice prevented air-sea gas exchange from occurring. In contrast, newly formed NADW was closer to equilibrium with the atmosphere because it had been transported northwards from the sub-tropics via surface currents before sinking in the high latitudes. With a larger difference between the $^{14}C$ ages and the water ages under glacial boundary conditions, the authors concluded that interpreting $^{14}C$ ages in terms of ventilation alone may lead to erroneous conclusions regarding ocean circulation and how it has changed in the past.*

- *Gebbie and Hubers (2012) noted that observational studies typically quantify the age of the deep Pacific Ocean to be between 700 and 1000 years, whereas most numerical models produce ages between 1500 and 2000 years. In an attempt to resolve this discrepancy, they used an inverse modelling framework to calculate the mean age of the deep Pacific from GLODAP $^{14}C$ observations. Their calculations suggested that deep North Pacific waters are between 1200 and 1500 years old. For comparison, the deep North Pacific water ages in FAMOUS are between 2000 and 2500 years, whereas the $^{14}C$ ages in this region are approximately 1750 years.*

- *Koeve et al. (2015) examined how slow and incomplete $^{14}CO_2$ air-sea gas exchange affects $\Delta^{14}C$ distributions in the interior ocean in the context of model evaluation. The authors defined the preformed age as the $^{14}C$ age of water relating to $^{14}C$ equilibration between the ocean and the atmosphere, which was very sensitive to air-sea gas exchange rates and moderately sensitive to sea ice cover. The bulk age of the water was defined as the age due to circulation plus the preformed age. The relative contribution of the preformed age to the bulk age was (on average) 50 %.*

*Miscellaneous*

Abstract, line 13: What do you mean by 'over-deep' NADW?

*The North Atlantic Deep Water cell in FAMOUS extends to a depth of ≈5 km, as opposed to ≈3.5 km, which is observed in the modern oceans. This will be rephrased for clarity in the revised manuscript.*

Page 5, line 9: "...post-bomb deep ocean $\Delta^{14}$C (i.e. natural $^{14}$C distributions)" I would not qualify the post-bomb deep radiocarbon as 'natural'; while some large areas might still hold the pristine signal many other deep ocean areas with younger age could be contaminated by the bomb signal.

*Given the (centennial-to-millennial) timescale of deep ocean ventilation compared to the time since the bomb contamination (which is a maximum of 50 years in our simulations), most parts of the deep ocean remain natural (aside from deep water formation regions, for example, where we can clearly see the bomb signal in our analyses). We will therefore add the caveat that post-bomb deep ocean $\Delta^{14}$C represents "mostly" natural $^{14}$C distributions to our revised manuscript.*

Pages 7 and 8, lines 14-18 and 24-26: these lines are unnecessary since the 'Abiotic $^{14}$C/$^{12}$C' flux (Eq. 6) does not call for the computation of aqueous $CO_2$.

*The lines in question are relevant to the air-sea gas exchange fluxes and were included for completeness. However, since we prefer to remove all references to the abiotic tracer scheme from the revised manuscript (in response to the comments about the interpretation of radiocarbon anomalies), these lines will also be removed.*

Page 7, equation 2: is the constant 100 or 1000?

*The constant is 100. Following Toggweiler et al. (1989):*

$$\delta^{14}C = (Model\ units - 100) \times 10$$

$$Model\ units = \frac{\delta^{14}C}{10} + 100$$

$$= \frac{\left(\dfrac{\dfrac{DI^{14}C}{DI^{12}C} - \left(\dfrac{^{14}C}{^{12}C}\right)_{std}}{\left(\dfrac{^{14}C}{^{12}C}\right)_{std}}\right) \times 1000}{10} + 100$$

$$= \frac{\left(\dfrac{\dfrac{DI^{14}C}{DI^{12}C} - \left(\dfrac{^{14}C}{^{12}C}\right)_{std}}{\left(\dfrac{^{14}C}{^{12}C}\right)_{std}}\right) \times 1000}{10} + 100$$

$$= \left(\frac{\dfrac{DI^{14}C}{DI^{12}C}}{\left(\dfrac{^{14}C}{^{12}C}\right)_{std}} - 1\right) \times 100 + 100$$

$$= \frac{DI^{14}C}{DI^{12}C} \times \frac{100}{^{14}C/_{^{12}C_{std}}}$$

Page 7, lines 17-18: "In the calculation of aqueous $CO_2$, we use the carbonic acid constants of Roy et al. (1993) as opposed to Millero (1995) because this is consistent with the formulation of $CO_2$ solubility used in other areas of the model." What is the rationale for requesting coherency between solubility and carbonate dissociation constants? These are two totally different topics. Shouldn't the dissociation constants be consistent among each others and with the pH scale used in the model?

*In response to comments about the interpretation of radiocarbon anomalies, we prefer to remove all references to the abiotic tracer from the revised manuscript, which includes these lines.*

Page 9, subsection title and line 16 (L term): does the transport only include advection in the model?
*As above, the tracer transport scheme computes changes in tracer concentrations due to advection, diffusion, and mixing. We will make the following changes in the revised manuscript to clarify this:*

- *Section 2.2.5 will be renamed "Transport" instead of "Advection"*
- *"L" will be redefined as the "transport term, which includes advection (using flux-limited Quadratic Upstream Interpolation for Convective Kinematics (QUICK)), diffusion, and mixing" instead of the "advection term".*

Section 3.3: why restrict the transient study to the North Atlantic? There are existing coral records in various other locations too (e.g. Druffel, 2002).
*We have chosen to focus on the North Atlantic because:*

- *The data are readily available online*
- *There is very good spatiotemporal coverage of the data (covering a wide range of depths in the sub-surface and intermediate ocean, and with adequate temporal coverage to capture the resolution and duration of the bomb spike), whereas this is not always the case in other regions*
- *We are interested in looking at if/how the bomb signal is recorded in natural archives at intermediate depths in comparison to the surface ocean (i.e. how the bomb signal penetrated through depth at a single location, and the maximum depth at which bomb $^{14}$C can be detected), and whether or not this is corroborated by the model. The North Atlantic data we compared to present a useful case study for this exercise, whereas the data presented by Druffel (2002) are typically located in the surface and sub-surface ocean, and therefore are not as pertinent to this particular study. However in publishing our $^{14}$C implementation and the example application in the present manuscript, we are facilitating similar analyses to be undertaken for various other locations (across different ocean basins and water depths) in the future.*

*We will clarify our reasons for focussing on the North Atlantic Ocean at the start of Section 3.3 in the revised manuscript.*

Page 16, lines 31-32: "We propose that this asymmetry relates to the age of the waters that are being upwelled in each basin." The 'we propose' formulation is confusing with respect to the correct explanation that follows.
*We will remove "we propose" in the revised manuscript.*

Page 19, lines 5-6: "However, Campin et al. (1999) did not account for isotopic fractionation in their study, nor was their $^{14}$C tracer cycled through the marine biological pump." This sentence is out of topic. Campin et al. (1999) represented the normalized $\Delta^{14}$C ratio in their model. Fractionation effects are therefore canceled and would not explain any difference in age behavior. Campin et al. (1999) did indeed not consider biological activity, but its impact is rather small on $\Delta^{14}$C contours (Bacastow and Maier-Reimer, 1990). Ages are computed with the help of $\Delta^{14}$C not with the $\delta^{14}$C ratios.
*We will remove this sentence from the revised manuscript.*

Article by Dentith et al. (submitted), quoted at different places in the text, is nowhere to be found on the editor (gmd) site. It would have been an advantage to have been able to consult it in order of understanding the methodology briefly presented in section 2.2.3.

*Technical journal requirements held up the progress of this manuscript, but it is now available in GMDD (https://www.geosci-model-dev-discuss.net/gmd-2019-250/0).*

*References*

Broecker WS, Gerard RD, Ewing M, Heezen BC. Geochemistry and physics of ocean circulation. In: Sears M, editor. Oceanography. Washington: AAAS. Publication 67, p 301-322, 1961.

Butzin, M., P. Köhler, and G. Lohmann, Marine radiocarbon reservoir age simulations for the past 50,000 years, Geophys. Res. Lett. 44, 8473–8480, doi:10.1002/2017GL074688, 2017.

Campin, J., Fichefet, T., Duplessy, J. Problems with using radiocarbon to infer ocean ventilation rates for past and present climates. Earth Planet. Sci. Lett. 165, 17–24, 1999.

Delhez E.J.M., E. Deleersnijder, A. Mouchet and J-M. Beckers. A note on the age of radioactive tracers, Journal of Marine Systems, 38, 277-286, 2003.

Druffel, E. R. M. Radiocarbon in corals: records of the carbon cycle, surface circulation and climate. Oceanography,15, 122–127. 2002.

Fiadeiro, M. E. Three-Dimensional Modeling of Tracers in the Deep Pacific Ocean, II. Radiocarbon and the Circulation. Journal of Marine Research, 40, 537–550, 1982.

Franke, J., M. Schulz, A. Paul, and J. F. Adkins. Assessing the ability of the 14C projection-age method to constrain the circulation of the past in a 3-D ocean model, Geochem. Geophys. Geosyst., 9, Q08003, doi:10.1029/2008GC001943, 2008.

Gebbie, G., and P. Huybers. The mean age of ocean waters inferred from radiocarbon observations: sensitivity to surface sources and accounting for mixing histories. J. Phys. Oceanogr., 42, doi: 10.1175/JPO-D-11-043.1, pp. 291–305, 2012.

Khatiwala, S., F. Primeau, and M. Holzer. Ventilation of the deep ocean constrained with tracer observations and implications for radiocarbon estimates of ideal mean age. Earth and Planet. Sci. Lett., doi:10.1016/j.epsl.2012.01.038, 2012.

Koeve, W., Wagner, H., Kähler, P., and Oschlies, A.: 14C-age tracers in global ocean circulation models, Geosci. Model Dev., 8, 2079–2094, https://doi.org/10.5194/gmd-8-2079-2015, 2015.

Maier-Reimer E. Geochemical cycles in an ocean general circulation model. Preindustrial tracer distributions. Global Biogeochemical Cycles 7: 645-677, 1993.

Mouchet A. The Ocean Bomb Radiocarbon Inventory Revisited. Radiocarbon 55: 1580-1594, 2013.

Orr, J. C., Najjar, R. G., Aumont, O., Bopp, L., Bullister, J. L., Danabasoglu, G., Doney, S. C., Dunne, J. P.,

Dutay, J.-C., Graven, H., Griffies, S. M., John, J. G., Joos, F., Levin, I., Lindsay, K., Matear, R. J., McKinley, G. A., Mouchet, A., Oschlies, A., Romanou, A., Schlitzer, R., Tagliabue, A., Tanhua, T., and Yool, A.: Biogeochemical protocols and diagnostics for the CMIP6 Ocean Model Intercomparison Project (OMIP), Geosci. Model Dev., 10, 2169–2199, https://doi.org/10.5194/gmd-10-2169-2017, 2017.

Orr, J. C., Najjar, R., Sabine, C. L., and Joos, F.: Abiotic-HOWTO, Internal OCMIP Report, LSCE/CEA Saclay, Gif-sur-Yvette, France, 25 pp., http://ocmip5.ipsl.jussieu.fr/OCMIP/phase2/simulations/Abiotic/HOWTO-Abiotic.html, 1999.

Toggweiler JR, Dixon K, Bryan K. Simulations of radiocarbon in a coarse-resolution world ocean model. 1. Steady state prebomb distributions. Journal of Geophysical Research 94:8217–42. 1989.

---

## Author Comment (AC1)

**Response to reviewer #1**

Dentith et al. describe the implementation of radiocarbon ($^{14}$C, or $\Delta^{14}$C when considering the normalized and fractionation-corrected $^{14}$C/$^{12}$C ratio) into the ocean component of the FAMOUS atmosphere-ocean GCM and present two $^{14}$C simulations, one spin-up simulation with constant pre-industrial boundary conditions plus a "historical" simulation forced with transient values of atmospheric $CO_2$ and $\Delta^{14}$C for 1765–2000 CE. The simulation results are compared with water column measurements available for the 1950s and the 1990s, and with $^{14}$C records from bivalves and deep-water corals spanning the period 1880–2000 CE. The model results are at least qualitatively in line with observations. FAMOUS simulates radiocarbon in two ways. The "abiotic" approach only considers uptake, transport and radioactive decay of normalized $^{14}$C/$^{12}$C. The "biotic" approach also considers isotopic fractionation due to air-sea gas exchange and the isotopic imprint of the biological pump. Comparing both approaches, Dentith et al. find that the "biotic" approach results in slightly elevated $^{14}$C/$^{12}$C ratios in the deep sea (by about 20 ‰ in terms of $\Delta^{14}$C). This result corroborates early findings stating that biological effects on $^{14}$C/$^{12}$C are much smaller (<10 %) than the effects of transport and radioactive decay (Fiadeiro, 1982). In addition, Dentith et al. compare simulated $^{14}$C water ages with "true" water ages according to an idealized age tracer. It turns out that for large parts of the oceans, a purely kinematic interpretation of $^{14}$C/$^{12}$C ages can lead to erroneous conclusions.

The paper is well written, the presentation of the results is mostly clear, and the literature is comprehensive. It may be published if the following issues are amended:

*We would like to thank reviewer #1 for their feedback on our manuscript. We have considered all of the comments carefully and addressed them in turn below, with our responses in blue-italics.*

**Major issue:** As described in Section 2.3.2, the model setup does not allow for meridional variability of atmospheric $\Delta^{14}$C. Therefore, the historical simulation misses the strong interhemispheric $\Delta^{14}$C gradient in the atmosphere of up to 400 ‰ (e.g., Fig. 1 in the manuscript) due to nuclear weapons testing. As a consequence, the model may underestimate the actual spatiotemporal $^{14}$C gradients in the ocean since the late 1950s. This shortcoming may distort the evolution of bomb $^{14}$C transients discussed in Section 3.3. It may also lead to biased post-bomb $^{14}$C distributions in subsurface and deeper waters which are discussed in Section 3.2. I would strongly recommend repeating the historical simulation forced with hemispheric averages of atmospheric $\Delta^{14}$C. The implementation should not be too difficult (the model has to read in three files provided by OCMIP-2).

*Whilst we agree that prescribing multiple latitude bands for both atmospheric $CO_2$ and $\Delta^{14}C$ would allow the north-south gradient in fossil fuel emissions and the location of atmospheric nuclear weapons testing to be better represented, this would require substantial changes to the code because FAMOUS, as with all configurations of version 4.5 of the Unified Model, currently only allows a single atmospheric value to be prescribed per year (both spatially and temporally). This project is no longer being funded, therefore we are unable to make this change.*

*In the future, it is intended that the isotope-enabled model will primarily be used to study changes in ocean circulation and the marine carbon cycle in a palaeo context, for example, at the Last Glacial Maximum (21,000 years ago) and during the last deglaciation (21,000 to 11,000 years ago). On these timescales, the atmosphere can be considered to be globally well-mixed, therefore meridional variability would not be required.*

**Further issues and comments (P = page, L = line, SM = supplementary material):**

P 3, L 6: The value of the half-life should be consistent with the updated value of 5700 years promoted in the SM.

*Although this would be a minor revision to the code, re-running the simulations and repeating the analyses would be a significant amount of work for minimal gain, especially since this project is no longer being funded. As outlined in the Supplementary Material, we recommend that the revised half-life is used for future scientific applications of the isotope-enabled model (i.e. the next time that a 10,000 year spin-up simulation is conducted, the new half-life should be employed as standard). However, we do not feel that it is necessary to make this change for our current study because the outdated half-life (5730 ±40 years) and revised half-life (5700 ±30 years) have overlapping error ranges, and re-running the model and our analyses with the updated half-life would not significantly alter the simulated isotope distributions or our interpretations.*

*The general formula for exponential decay is:*

$$N_{(t)} = N_0 2^{-\frac{t}{t_{1/2}}}$$

*where $N_{(t)}$ is the amount of substance remaining at time 't', $N_0$ is the initial amount of the substance, and $t_{1/2}$ is the half-life of the substance.*

*We can therefore calculate the error associated with using the outdated half-life. At the end of the spin-up simulation (i.e. when $t = 10,000$), the amount of $^{14}C$ remaining ($N_{10,000}$) using the outdated half-life of 5730 years is 0.298, whereas the amount remaining with the revised half-life of 5700 years is 0.296. This is the maximum error because the difference between the two half-lives is smaller for younger water parcels. For example, $N_{1000}$ is 0.8855 using a half-life of 5730 years, and 0.8861 using a half-life of 5700 years.*

P 3, L 28: "Indian" should read "Indien"
*This typographical error will be corrected in the revised manuscript.*

P 4, L 28: The models MOM (Toggweiler et al., 1989), LOCH (Mouchet, 2013) and LOVEVLIM (e.g., Menviel et al. 2017) could be added to the list.
*Our intention was to provide illustrative examples of $^{14}C$-enabled models across a range of complexities as opposed to a complete list of all $^{14}C$-enabled models, but we are happy to add these additional examples in the revised manuscript.*

P 7, L 28: (660 / Sc)**-0.5 should read (Sc / 660)**-0.5 (e.g., Orr et al. 2017, equation (12))
*This typographical error will be corrected in the revised manuscript.*

P 7, L 31: Missing reference (I guess it is Wanninkhof 1992)
*The reference (Wanninkhof, 1992) will be added in the revised manuscript.*

P 8, L 8 and P 22, L 27: I could not find the paper at GMDD. What is the current status of the manuscript?
*Technical journal requirements held up the progress of this manuscript, but it is now available in GMDD (https://www.geosci-model-dev-discuss.net/gmd-2019-250/0).*

P 8, L 12: 14 and 13 should be subscripts

*These typographical errors will be corrected in the revised manuscript.*

P 8, L 22: "δ13C is close to zero": this is the only case for DIC but not for phytoplankton

*We will amend the text to "$\delta^{13}C_{DIC}$ is close to zero" to avoid confusion.*

P 9, L 10: This is not correct, Stuiver and Pollach (1977) employ 5568 years.

*Stuiver and Polach (1977) is the reference for the equation:*

$$^{14}C_{age} = -\frac{\lambda}{\ln(2)} \times \ln\left(1 + \frac{\Delta^{14}C}{1000}\right).$$

*However, we will add a sentence in the revised manuscript to clarify that we employ the Cambridge half-life ($\lambda = 5730$ years), instead of the Libby half-life ($\lambda = 5568$ years) used by Stuiver and Polach (1977).*

P 9, L 16: What about turbulent diffusion, where is it included?

*The ocean model has a tracer transport scheme that computes changes in tracer concentrations due to advection, diffusion, and mixing. We will make the following changes in the revised manuscript to clarify this:*

- *Section 2.2.5 will be renamed "Transport" instead of "Advection"*
- *"L" will be redefined as the "transport term, which includes advection (using flux-limited Quadratic Upstream Interpolation for Convective Kinematics, QUICK), diffusion, and mixing" instead of the "advection term".*

P 9, L 26–28 and Figure S1: The OCMIP-2 steady state criterion is somewhat different, demanding that 98% of the ocean volume has a $\Delta^{14}C$ drift of less than 0.001 ‰ per year (see also Aumont et al. 1998). The total $^{14}C$ inventory may have stabilized while there might be still an ongoing internal redistribution of $^{14}C$. Have you checked this?

*We have verified that the internal redistribution of $^{14}C$ at the end of the spin-up simulation is minimal. We will revise the final sentence of Section 2.3.1 as follows:*

*"At the end of the spin-up simulation, ≈99.8 % of the ocean has a $\Delta^{14}C$ drift of less than 0.001 ‰ yr$^{-1}$ (equivalent to a change in $^{14}C$ age of 8.27 years per millennia), satisfying the OCMIP-2 criterion for steady state (Orr et al., 2000)."*

P 13, L 5–6: "(...) the simulated $\Delta^{14}C$ gradient between the surface ocean and approximately 1000 m depth is shallower than observed (. . .). This is visible in all regions, except the Northern Hemisphere deep water formation region (NH_DWF) and the North Pacific (NP) (. . .)." According to Figure 8 the opposite is true (indicating steeper simulated gradients everywhere except for NH_DW and NP).

*In the near-surface ocean, the simulated change in $\Delta^{14}C$ per unit depth is less than the observed change in all regions except the Northern Hemisphere deep water formation region and the North Pacific. We will correct this sentence in the revised manuscript, replacing "shallower" with "steeper".*

P 13, L 23: "(. . .) due to the sparsity of data." You might wish to compare your results with post-bomb $\Delta^{14}C$ (bottle) data provided by GLODAPv2 (https://odv.awi.de/data/ocean/glodap-v22019-bottle-data/).

*We have downloaded the GLODAPv2 Arctic Ocean data set from: https://www.glodap.info/index.php/merged-and-adjusted-data-product/. However, we have only been*

*able to identify 64 $\Delta^{14}C$ data points from the 1990s. Specifically, these data are from six depth transects (with between 9 and 16 measurements at each location). They therefore provide insufficient resolution to contribute to our original spatial comparisons (Figures 4 and 6 in the original manuscript). However, a simple linear regression between these data and our simulated values at the corresponding grid cells in the model demonstrates that our simulated values are consistently lower than observed (Figure R1), which we propose is largely because FAMOUS has a cold temperature bias in the Arctic Ocean and around the coast of Greenland and Iceland that is linked with expanded annual sea ice in the Nordic and Labrador Seas (Dentith et al., 2019 - [https://link.springer.com/article/10.1007/s00382-018-4243-y](https://link.springer.com/article/10.1007/s00382-018-4243-y)) – i.e. limiting the input of bomb $^{14}C$ from the atmosphere to the ocean in this region. The observed depth transects extend between the surface ocean and approximately 3000 m depth, which approximately corresponds to the depths over which our simulated globally-averaged depth profile has a negative $\Delta^{14}C$ bias relative to the observed profile (Figure 8 in the original manuscript). Thus, as well as our simulated global depth profile including the relatively low Arctic Ocean values that are missing from the original gridded GLODAP data set, our $\Delta^{14}C$ values are lower than observed in this region, further compounding the negative bias. We will add a brief discussion of this analysis to the revised manuscript after these original sentences: "The masking in the GLODAP data set also contributes towards some of the offset between the model and the observations. For example, we include the relatively low Arctic Ocean values in our global, Northern Hemisphere deep water formation region (NH_DWF) and Labrador Sea (LS) profiles, but these latitudes are masked out in GLODAP due to the sparsity of data (Figure 4a)".*

[Figure]

***Figure R1:** Simulated versus observed $\Delta^{14}C$ values from the Arctic Ocean during the 1990s.*

P 17, L 19: This is an interesting result supporting and quantifying the notion by Fiadeiro (1982) that $^{14}C/^{12}C$ can roughly be regarded as radio-conservative tracer in the deep sea.

*We also thought this was an interesting result. However, in response to feedback from reviewer #2, we have decided to remove the abiotic-biotic comparison from our revised manuscript. This is because we have not yet found a way to present these results to both biogeochemical modellers and analytical biogeochemists without causing contention and confusion (see our response to reviewer #2 for further details). In the revised manuscript, we will remove all references to the pre-existing abiotic tracer, and instead will only present the comparison between biotic $\Delta^{14}C$ and $\Delta^{14}C$ observations, and the biotic $^{14}C$ ages and the simulated water ages.*

P 18, L 1–6: The simulated $^{14}C$ ages reflect local ageing, transport, biogeochemistry, and radioactive decay. On the other hand, the simulated ideal water ages only reflect local ageing and transport. As there is no radioactive decay of the age tracer, the age differences cannot be entirely explained with biogeochemical effects.

*Here, we consider the ageing of water and the radioactive decay of particles within a water mass to be equivalent processes – both are affected by the time since being reset at the surface. For example, because radioactive decay occurs at a known rate through time, this can be converted to an age since being reset at the surface. In Figure S6, we take into account that the water age is instantaneously reset at the surface, whereas the $^{14}C$ has a reservoir age associated with isotopic equilibration rates, sea ice cover, and mixing, by normalising each depth profile relative to the surface ocean in each region; we therefore think it is a reasonable approximation to attribute the differences between the water ages and the $^{14}C$ ages to ocean biogeochemistry.*

P 18, L 15–16: "(...) the water ages generally increase with depth because they are a simple function of advection." I disagree, there may be considerable mixing leading to nonlinear ageing of water parcels. This is seen in many tracers records.

*In the revised manuscript, we will clarify that the water ages generally increase with depth because they are a function of the large-scale transport (i.e. in general, water parcels in the deep ocean have been out of contact with the atmosphere longer than water parcels at intermediate depths). We will also clarify that the increase in water age with depth is non-linear because of the mixing of water parcels of different ages.*

P 31, Figure 4 (b): I would prefer to see the differences between simulated and observed values, analogously to Figure 2.

*We will add a third subplot to Figure 4 to show the difference between the simulated and observed values, as below:*

[Figure]

*Figure 4:* *(c) Simulated minus observed Δ¹⁴C values.*

Figures 8, 10, 11, 16, S4, S5, and S6: Lines and dots are blurred.

*There was no option during the original manuscript submission to submit the high resolution PDF figures. These will be uploaded for the production of the final revised paper. In the interim period, we would be happy to supply the editor with a zip file of the original PDFs to be shared with the reviewers. Please let us know what is preferred.*

P 38, Figure 11: The line plots should be replaced with Hovmöller diagrams.

*We will replace the original line plots with Hovmöller diagrams and revise the figure caption, as below:*

[Figure]

**Figure 11:** *Hovmöller diagrams of simulated $\Delta^{14}C$ at the coral and bivalve locations (outlined in Figure 9): Bermuda (B), Bay of Biscay (BB), Grimsey (G), German Bight (GB), Georges Bank (GeB), Grand Banks (GrB), Hudson Strait (HS), Northeast Channel (NE), Oyster Ground (OG), Siglufjörður (S), Sable Bank (SB), and Tromsø (T). Note that the vertical scale has been expanded for the uppermost 100 m of the water column.*

SM, P 2: "OCMIP" should read "OCMIP-2"
*This typographical error will be corrected in the revised manuscript.*

**References:**

Aumont, O.; Orr, J. C.; Jamous, D.; Monfray, P.; Marti, O.; Madec, G. A Degradation Approach to Accelerate Simulations to Steady-State in a 3-D Tracer Transport Model of the Global Ocean. Climate Dynamics 1998, 14 (2), 101–116.

Fiadeiro, M. E. Three-Dimensional Modeling of Tracers in the Deep Pacific Ocean, II. Radiocarbon and the Circulation. Journal of Marine Research 1982, 40, 537–550.

Menviel, L.; Yu, J.; Joos, F.; Mouchet, A.; Meissner, K. J.; England, M. H. Poorly Ventilated Deep Ocean at the Last Glacial Maximum Inferred from Carbon Isotopes: A Data-Model Comparison Study. Paleoceanography 2017, 32 (1), 2016PA003024.

Mouchet, A. The Ocean Bomb Radiocarbon Inventory Revisited. Radiocarbon 2013, 55, 1580–1594.

Orr, J. C.; Najjar, R. G.; Aumont, O.; Bopp, L.; Bullister, J. L.; Danabasoglu, G.; Doney, S. C.; Dunne, J. P.; Dutay, J.-C.; Graven, H.; et al. Biogeochemical Protocols and Diagnostics for the CMIP6 Ocean Model Intercomparison Project (OMIP). Geosci. Model Dev. 2017, 10 (6), 2169–2199.

Orr, J. C.; Najjar, R. G.; Aumont, O.; Bopp, L.; Bullister, J. L.; Danabasoglu, G.; Doney, S. C.; Dunne, J. P.; Dutay, J.-C.; Graven, H.; et al. Biogeochemical Protocols and Diagnostics for the CMIP6 Ocean Model Intercomparison Project (OMIP). Geosci. Model Dev. 2017, 10 (6), 2169–2199.